# CoCon: A Self-Supervised Approach for Controlled Text Generation

**Alvin Chan**[1]*, **Yew-Soon Ong**[1], **Bill Pung**[1], **Aston Zhang**[2], **Jie Fu**[3]
[1]Nanyang Technological University,     [2]Amazon AI,     [3]Mila, Polytechnique Montreal

## Abstract

Pretrained Transformer-based language models (LMs) display remarkable natural language generation capabilities. With their immense potential, controlling text generation of such LMs is getting attention. While there are studies that seek to control high-level attributes (such as sentiment and topic) of generated text, there is still a lack of more precise control over its content at the word- and phrase-level. Here, we propose Content-Conditioner (CoCon) to control an LM's output text with a content input, at a fine-grained level. In our self-supervised approach, the CoCon block learns to help the LM complete a partially-observed text sequence by conditioning with content inputs that are withheld from the LM. Through experiments, we show that CoCon can naturally incorporate target content into generated texts and control high-level text attributes in a zero-shot manner.[1]

## 1 Introduction

Transformer-based (Vaswani et al., 2017; Tay et al., 2020) pretrained language models (LMs) have led a wave of new advances in natural language processing tasks as a means to extract contextualized word embeddings (Devlin et al., 2018; Dai et al., 2019b; Yang et al., 2019) and as text generators (Radford et al., 2019; Brown et al., 2020). These LMs are trained on huge amounts of text corpora to predict next tokens through a log-likelihood objective. Given its remarkably fluent text generation, there is growing interest in controlling output texts of such LMs (Keskar et al., 2019; Dathathri et al., 2019). Approaches like training a modified LM from scratch to incorporate target text attributes (Keskar et al., 2019) can be expensive while finetuning pretrained LMs for specific attributes (Ziegler et al., 2019) limits the scope of text control. Without changing the architecture or weights of pretrained LMs, one promising approach (PPLM) (Dathathri et al., 2019) controls generated text through attribute models. Though effective in controlling high-level text attributes such as topic and sentiment, the same target attribute may generate text samples with vastly different content at the word- and phrase-levels, leaving a gap for more fine-grained control over the content of LM-generated texts.

We conceptualize Content-Conditioner (CoCon) as an approach to narrow this gap by guiding pretrained LMs' text outputs through the incorporation of content input. This content input can take the form of a text sequence whose content we would like to condition on for text generation. Essentially, CoCon comprises two parts: 1) a pretrained LM and 2) a interleave CoCon layer. By employing a pretrained LM, CoCon incorporates the representations of a content input into the encoded text representations through the CoCon layer before passing the content-conditioned representations into $LM_\beta$ for generation. To train the CoCon block, we propose a self-supervised learning approach where training data consist of text samples generated by the pretrained LM itself (§ 3.1). By splitting each text sequence into two segments ($[\mathbf{x}^a; \mathbf{x}^b]$), CoCon learns through a self reconstruction objective to help the LM reconstruct missing latter segments ($\mathbf{x}^b$) by taking $\mathbf{x}^b$ itself as the content input. We use content masking for CoCon and also propose other loss functions such as cycle reconstruction to condition content from divergent sources while producing high-quality texts. Since the CoCon block's size is a small fraction of the LM and no finetuning is conducted on the LM's weights, the training cost is significantly lower than training an LM from scratch. We show that CoCon's fine-grained content control can be extended to also influence higher-level text attributes

---

*Corresponding author: `guoweial001@ntu.edu.sg`
[1]Codes and models are available at: `https://github.com/alvinchangw/COCON_ICLR2021`

such as topic and sentiment in a zero-shot manner, and compare it with strong controlled generation baselines. Furthermore, CoCon is versatile in assimilating multiple content inputs, and its strength of content-conditioning can be flexibly adjusted through a content bias term during inference. In this paper, we demonstrate the CoCon approach with the GPT-2 345M model (Radford et al., 2019) as the pretrained LM. Given CoCon's modular nature, it can be used with other Transformer-based LMs or even other controlled generation methods. All in all, the core contributions of this paper are:

- We propose CoCon for content-conditioned language generation.

- We introduce a self-supervised learning approach where CoCon learns to complete text sequences when given information about future tokens.

- Through ablation studies and comparisons with strong baselines like PPLM and CTRL (Keskar et al., 2019), we investigate how CoCon controls high-level attributes such as topic and sentiment while generating texts that have high content similarity to conditioning text.

## 2 RELATED WORK

There is a line of work that aims to generate output text of desired attributes with neural networks. Some of the earliest efforts involve conditional generative models (Kikuchi et al., 2016; Ficler & Goldberg, 2017) where the networks are trained on text data labeled with the target attributes. These models can be trained via reinforcement learning (Ziegler et al., 2019) or the generative adversarial network (Yu et al., 2017) framework. Unlike CoCon, the requirement of predetermined attributes in those methods limits the possible types of generated texts. CTRL (Keskar et al., 2019) is a recent approach that generated controlled fluent texts through the use of control codes which are meta-data prepended to the text during generation. Though it produces high-quality text with its GPT-2-like architecture, its control codes are also predetermined during the training. Closest to our work is Plug and Play Language Model (PPLM) (Dathathri et al., 2019) which seeks to control text on already pretrained LM without finetuning through relatively small 'pluggable' attribute models. While PPLM's flexible design also enables controlled generation without retraining or finetuning the LM like in CoCon, our approach aims to control the generation at a content level, beyond high-level text attributes. Another core difference lies in the training where CoCon's self-supervised learning absolves the need for labeled data, such as the ones employed to train PPLM's attribute discriminator models. Weighted decoding (Ghazvininejad et al., 2017; Holtzman et al., 2018) seeks to control the output text token by upweighting the probabilities of targeted words during the decoding step but has been shown to produce incoherent text (See et al., 2019). Conditioning language generation has been used in question generation to enhance faithfulness by attending to textual context such as predicates, subject types or object types (Elsahar et al., 2018) rather than the content input used here in CoCon. Small adapter layers (Bapna et al., 2019) have been previously proposed for multilingual translation to also save on model size and training resources but differ from CoCon's self-supervised training as they rely on annotated sentence pairs of different languages for training.

Text style transfer is a related area that controls texts' attributes by translating text from one style to another (Dai et al., 2019a). A few of such studies employ auto-encoders to separate texts' style and non-style latent representation (Shen et al., 2017; Hu et al., 2017; Yang et al., 2018). This disentanglement enables style changes to the text at the latent space while retaining most of its content. Another work identifies attribute markers (Li et al., 2018) which are $n$-grams correlated with a particular style in a text corpus and edit texts' style by substituting them. Essentially, style transfer alters existing texts rather than generating texts and requires predefined attributes.

## 3 CONTENT CONDITIONER (COCON)

In the following sections, we discuss the motivation for CoCon, its model architecture and how we train the CoCon block.

**Motivation** In text generation with language models, given the prompt text $x_{:t-1} = \{x_1, \ldots, x_{t-1}\}$, the following text $\{x_t, \ldots, x_l\}$ is generated in an auto-regressive manner (Man-

ning et al., 1999; Bengio et al., 2003):

$$p(x_t, \ldots, x_l | x_1, \ldots, x_{t-1}) = \prod_{i=t}^{l} p(x_i | x_1, \ldots, x_{i-1}). \tag{1}$$

Previous studies on controlled text generation in LM showed that $p(\mathbf{x})$ can be conditioned on target attributes (Dathathri et al., 2019) or control codes (Keskar et al., 2019) to control the text's sentiment or topic, i.e.,

$$p(x_t, \ldots, x_l | x_1, \ldots, x_{t-1}) = \prod_{i=1}^{l} p(x_i | \mathbf{a}, \{x_1, \ldots, x_{i-1}\}), \tag{2}$$

where $\mathbf{a}$ is the target attribute. While these methods show that the generation is fluent and can be aligned with the target attribute well, the output texts $\{x_t, \ldots, x_l\}$ are controlled at a global attribute (e.g., sentiment/topic) level rather than at a more local *content* (e.g., words/phrases) level. Since there is a vast number of possible $\{x_t, \ldots, x_l\}$ candidates which would align well with both the prompt text and target attribute, this results in generated text samples that contain very different content during the stochastic token sampling process. This motivates an approach to condition on an content input $\mathbf{c}$ for more fine-grained control over text generation:

$$p(x_t, \ldots, x_l | x_1, \ldots, x_{t-1}) = \prod_{i=1}^{l} p(x_i | \mathbf{c}, \{x_1, \ldots, x_{i-1}\}), \tag{3}$$

where $\mathbf{c}$ can be a text sequence whose content we would like to condition on during text generation. Next, we propose the model architecture of Content-Conditioner (CoCon) as an approach for this control.

**Model Architecture** Our proposed Content-Conditioner (Figure 1) controls the content of the generated text while maintaining fluency by incorporating a pretrained Transformer-based language model (LM), GPT-2 (Radford et al., 2019) in our experiments. Such LMs have shown remarkable natural text generation in the auto-regressive manner (Eq. 1) where the next token $x_t$ is sampled based on the logits $\mathbf{o}_t = \text{LM}(x_{:t-1})$. These LMs are essentially stacks of Transformer blocks, each consisting of layer normalization (Ba et al., 2016), multi-head self-attention (Vaswani et al., 2017) and position-wise feed forward operations.

An LM's generation can be broken down into two separate parts: layers before the CoCon block ($\text{LM}_\alpha$) and layers after ($\text{LM}_\beta$). The $\text{LM}_\alpha$ acts as a feature extractor that takes in the input sequence's embeddings and outputs its intermediate representation at a breakpoint, i.e., $\mathbf{h}_{:t-1} = \text{LM}_\alpha(x_{:t-1})$. Subsequently, $\text{LM}_\beta$ takes in this representation and outputs the logits for the next token, i.e., $\mathbf{o}_t = \text{LM}_\beta(\mathbf{h}_{:t-1})$, yielding

$$\mathbf{o}_t = \text{LM}(x_{:t-1}) = \text{LM}_\beta(\text{LM}_\alpha(x_{:t-1})) = \text{LM}_\beta(\mathbf{h}_{:t-1}). \tag{4}$$

From Eq. 4, we can see that the representation ($\mathbf{h}$) is a medium to control next token logits ($\mathbf{o}$) and hence the text generation process. Indeed, we transform $\mathbf{h}$ by conditioning it with the content input ($\mathbf{c}$) through a CoCon block such that

$$\mathbf{h}'_{:t-1} = \text{CoCon}(\mathbf{h}^{(\mathbf{c})}_{:l_c}, \mathbf{h}_{:t-1}), \tag{5}$$

where $\mathbf{h}^{(\mathbf{c})}_{:l_c} = \text{LM}_\alpha(\mathbf{c})$ is the content representations and $l_c$ is the length of the content text sequence. We parameterize the CoCon block as a single Transformer block with an attention and position-wise feed-forward operation. Similar to a typical LM attention layer, the query ($\mathbf{Q}$), key ($\mathbf{K}$), value ($\mathbf{V}$) matrices are computed through linear transformations on the representations $\mathbf{h}_{:t-1}$, where $\mathbf{Q}, \mathbf{K}, \mathbf{V} \in \mathbb{R}^{(t-1) \times d}$ and $d$ is the representations' dimension. To attend to the content

representations ($\mathbf{h}^{(\mathbf{c})}_{:l_c}$), the content keys and values ($\mathbf{K}^{(\mathbf{c})}, \mathbf{V}^{(\mathbf{c})} \in \mathbb{R}^{l_c \times d}$) are also computed, and concatenated to the original attention matrices before computing the CoCon attention output:

$$\mathbf{K}' = [\mathbf{K}^{(\mathbf{c})}; \mathbf{K}], \quad \mathbf{V}' = [\mathbf{V}^{(\mathbf{c})}; \mathbf{V}], \quad \mathbf{A} = \text{Softmax}(\mathbf{Q}\mathbf{K}'^{\top})\mathbf{V}' = \text{Softmax}(\mathbf{W})\mathbf{V}', \quad (6)$$

where $\mathbf{A} = \{\mathbf{a}_1, \ldots, \mathbf{a}_{t-1}\}$ and $\mathbf{W} \in \mathbb{R}^{(t-1) \times (l_c + t - 1)}$ represents the attention weights. The final CoCon outputs are computed with a position-wise feed-forward layer. By concatenating to the representations prior to $t - 1$ and passing them to $\text{LM}_{\beta}$, the next logits, and consequently word token $\tilde{\mathbf{x}}_t$, is now conditioned on $\mathbf{c}$:

$$\mathbf{h}'_i = \text{FF}(\mathbf{a}_i), \quad \tilde{\mathbf{o}}_t = \text{LM}_{\beta}([\mathbf{h}_{:t-2}; \mathbf{h}'_{-1}]), \quad p_{\theta,\psi}(\tilde{x}_t | \mathbf{c}, x_{:t-1}) = \text{Softmax}(\tilde{\mathbf{o}}_t), \quad (7)$$

where $\theta$ and $\psi$ are the paramterization of the CoCon block and LM respectively. Similar to a GPT-2 Transformer block, our CoCon block includes layer normalization before its multi-headed attention and feed-forward layers. Figure 1 summarizes the CoCon architecture which enables auto-regressive text generation by using $\tilde{x}_i$ as the token input ($x_i$) to generate $\tilde{x}_{i+1}$ where $i \geq t$.

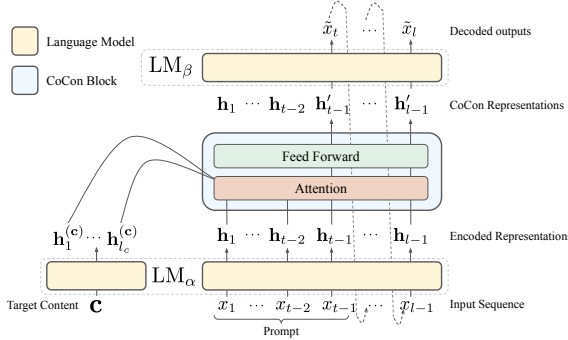

Figure 1: Model architecture of proposed Content-Conditioner (CoCon).

**Multiple Content Inputs** CoCon's flexible design enables *multiple* content inputs for a single generation. In the case where we have $N$ content inputs ($\mathbf{c}^1, \ldots, \mathbf{c}^N$), the output text can be conditioned by these contents through their attention keys and values, similar to Eq. 6:

$$\mathbf{K}' = [\mathbf{K}^{(\mathbf{c}^1)} \ldots \mathbf{K}^{(\mathbf{c}^N)}; \mathbf{K}], \quad \mathbf{V}' = [\mathbf{V}^{(\mathbf{c}^1)} \ldots \mathbf{V}^{(\mathbf{c}^N)}; \mathbf{V}], \quad \mathbf{A} = \text{Softmax}(\mathbf{Q}\mathbf{K}'^{\top})\mathbf{V}'. \quad (8)$$

**Strength of Content Conditioning** Within CoCon's attention mechanism, we can vary the extent of content conditioning on the output text by biasing the attention weights in $\mathbf{W}$ (Eq. 6) that correspond to the content input ($\mathbf{c}$). More specifically, the influence of $\mathbf{c}$ on the output text can be altered through the attention's softmax weighting on the content values ($\mathbf{V}^{(\mathbf{c})}$). During generation, a positive bias term ($\tau_{\text{content}}$) can optionally be added to the content attention weights $\mathbf{W}_{:,:l_c} \in \mathbb{R}^{(t-1) \times l_c}$ to increase influence of $\mathbf{V}^{(\mathbf{c})}$, boosting content conditioning, while a negative term can conversely reduce the content-conditioning effect. We discuss examples of varying $\tau_{\text{content}}$ in § 4.4.

## 3.1 SELF-SUPERVISED LEARNING

We train CoCon with a self-supervised learning approach that is inspired by the diversity of content in natural language. Given a text sequence $\mathbf{x} = \{x_1, \ldots, x_{t-1}, x_t, \ldots, x_l\}$ of length $l$, we can break it into two contiguous segments: $\mathbf{x}^a = \{x_1, \ldots, x_{t-1}\}$ and $\mathbf{x}^b = \{x_t, \ldots, x_l\}$ where $\mathbf{x} = [\mathbf{x}^a; \mathbf{x}^b]$. In the real world, there may be numerous substitutes of $\mathbf{x}^b$ that could follow from $\mathbf{x}^a$ fluently. Coupled with the randomness in text sampling, this means that, without information about $\mathbf{x}^b$, the probability of reconstructing the full $\mathbf{x}$ from $\mathbf{x}^a$ alone with an LM can be low.

**Self Reconstruction Loss** Based on this intuition, our approach trains the CoCon block to help the LM reconstruct the original $\mathbf{x}$ by also conditioning with $\mathbf{x}^b$ as the content input, i.e., $\mathbf{c} = \mathbf{x}^b$ (Figure 2b). More concretely, we first compute the intermediate representations of the input text $\mathbf{x}$ and $\mathbf{c}$:

$$\mathbf{h}_{:l} = \mathrm{LM}_\alpha(\mathbf{x}) = \mathrm{LM}_\alpha(x_{:l}), \quad \mathbf{h}_{:l_c}^{(\mathbf{c})} = \mathrm{LM}_\alpha(\mathbf{c}) = \mathrm{LM}_\alpha(x_{t:l}), \tag{9}$$

where $l_c = l - t + 1$ is the length of $\mathbf{c}$. The content-conditioned representation can be computed by the CoCon block where $\mathbf{h}_{:l_c}^{(\mathbf{c})}$ is the content representation:

$$\mathbf{h}_i' = \mathrm{CoCon}(\mathbf{h}_{:l_c}^{(\mathbf{c})}, \ \mathbf{h}_{:i}) \ , \quad \forall i \geq t - 1. \tag{10}$$

Similar to Eq. 7, the CoCon transformed representations are concatenated to the original representation before $t - 1$ and passed into $\mathrm{LM}_\beta$ to produce the LM logits:

$$\tilde{\mathbf{o}}_{i+1} = \mathrm{LM}_\beta([\mathbf{h}_{:t-2}; \mathbf{h}_{t-1:i}']), \quad p_{\theta,\psi}(\tilde{x}_{i+1}|\mathbf{c}, x_{:i}) = \mathrm{Softmax}(\tilde{\mathbf{o}}_{i+1}), \quad \forall i \geq t - 1. \tag{11}$$

Through an LM training objective, we arrive at the self-reconstruction loss term which trains CoCon to predict tokens of $\mathbf{x}^b$ by conditioning on $\mathbf{x}^b$ itself as the content input ($\mathbf{c}$):

$$\mathcal{L}_{\mathrm{self}} = -\sum_{i=t}^{l} \log p_{\theta,\psi}\left(x_i|(\mathbf{c} = \mathbf{x}^b), \{x_1, \ldots, x_{i-1}\}\right). \tag{12}$$

To avoid trivializing the prediction of the next token $x_{i+1}$ during training, we apply a self-token $\mathbf{c}$-mask at CoCon's attention layer such that $\mathbf{h}_i'$ does not attend to the token $x_{i+1}$ in $\mathbf{c}$ that it is trying to predict. This approach can be conducted in a self-supervised manner with any pretrained LM where the training samples $\mathbf{x}$ are generated text outputs stochastically sampled from the LM itself.

**Null Content Loss**  To encourage CoCon's outputs to follow the prompt text $\mathbf{x}^a$ fluently without relying on $\mathbf{x}^b$, we also train CoCon with a loss term similar to Eq. 12 but replaces the content input with a null token ($\varnothing$):

$$\mathcal{L}_{\mathrm{null}} = -\sum_{i=t}^{l} \log p_{\theta,\psi}\left(x_i|(\mathbf{c} = \varnothing), \{x_1, \ldots, x_{i-1}\}\right). \tag{13}$$

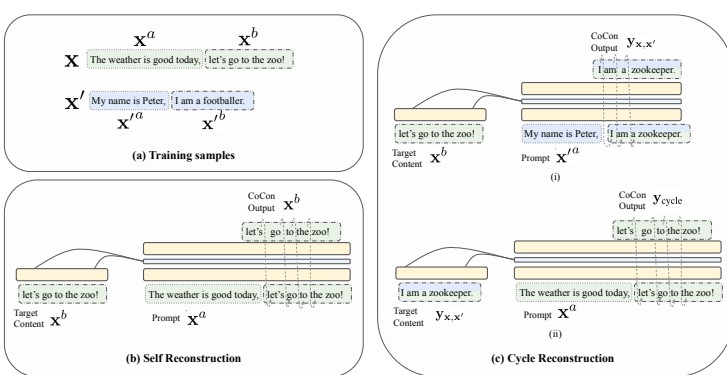

Figure 2: Illustrative examples of (b) self reconstruction and (c) cycle reconstruction training.

**Cycle Reconstruction Loss**  The self reconstruction loss relies on CoCon content input ($\mathbf{c}$) and initial prompt text ($\mathbf{p}$) originating from one single text sample. To encourage generalization on cases where $\mathbf{c}$ and $\mathbf{p}$ are from divergent text sources, we employ a cycle reconstruction training that utilizes two different training samples (e.g., $\mathbf{x}, \mathbf{x}'$ in Figure 2a) and two CoCon forward steps (Figure 2c). We can express the output of a CoCon's auto-regressive generation as

$$\mathbf{y} = f_{\theta,\psi}(\mathbf{c}, \mathbf{p}), \tag{14}$$

where $[\mathbf{p}; \mathbf{y}]$ would be a fluent text sequence and $\mathbf{y}$ is conditioned on the content of $\mathbf{c}$. The first step (Figure 2c(i)) computes the CoCon output with the content input ($\mathbf{c}$) sourced from $\mathbf{x}$ and prompt text ($\mathbf{p}$) sourced from $\mathbf{x}'$:

$$\mathbf{y}_{\mathbf{x},\mathbf{x}'} = f_{\theta,\psi}((\mathbf{c} = \mathbf{x}^b), (\mathbf{p} = \mathbf{x}'^a)), \tag{15}$$

where $\mathbf{x} = [\mathbf{x}^a; \mathbf{x}^b]$ and $\mathbf{x}' = [\mathbf{x}'^a; \mathbf{x}'^b]$. Since CoCon utilizes a pretrained LM for generation, $\mathbf{y}_{\mathbf{x},\mathbf{x}'}$ would be a text sequence that fluently follows the prompt, $\mathbf{x}'^a$, while seeking to incorporate $\mathbf{x}^b$'s content. The second CoCon forward step (Figure 2c(ii)) takes $\mathbf{y}_{\mathbf{x},\mathbf{x}'}$ as content input and $\mathbf{x}^a$ as prompt text:

$$\mathbf{y}_{\text{cycle}} = f_{\theta,\psi}((\mathbf{c} = \mathbf{y}_{\mathbf{x},\mathbf{x}'}), (\mathbf{p} = \mathbf{x}^a)), \tag{16}$$

Since $\mathbf{x} = [\mathbf{x}^a; \mathbf{x}^b]$, $\mathbf{x}^b$ is a valid continuation from the prompt $\mathbf{x}^a$ and recall that $\mathbf{y}_{\mathbf{x},\mathbf{x}'}$ was content-conditioned on $\mathbf{x}^b$ in the first CoCon step (Eq. 15). This posits $\mathbf{x}^b$ as a training label for $\mathbf{y}_{\text{cycle}}$ which gives us the cycle reconstruction loss term:

$$\mathcal{L}_{\text{cycle}} = -\sum_{i=t}^{l} \log p_{\theta,\psi}\left(\mathbf{y}_{\text{cycle}} = \mathbf{x}^b | (\mathbf{c} = \mathbf{y}_{\mathbf{x},\mathbf{x}'}), (\mathbf{p} = \mathbf{x}^a)\right). \tag{17}$$

**Adversarial Loss**   Adversarial training objectives have shown to help in generating realistic text outputs (Yang et al., 2018). Here, we also employ an adversarial training loss (Goodfellow et al., 2014) to encourage the output texts' representations ($\text{LM}_\alpha(\mathbf{y})$) to match those of the training samples ($\text{LM}_\alpha(\mathbf{x})$) by minimizing the loss:

$$\mathcal{L}_{\text{adv}} = \mathbb{E}_{\mathbf{x}}[\log f_{\text{disc}}(\text{LM}_\alpha(\mathbf{x}))] + \mathbb{E}_{\mathbf{y}}[\log(1 - f_{\text{disc}}(\text{LM}_\alpha(\mathbf{y})))], \tag{18}$$

where $f_{\text{disc}}$ is a discriminator network that classifies whether the representations are of CoCon-generated texts. Through continuous approximation of discrete sampling of $y$ where token logits instead of one-hot vectors are fed as input into $\text{LM}_\alpha$, CoCon and $f_{\text{disc}}$ can be trained with back-propagation in an end-to-end manner. Parameterizing the $f_{\text{disc}}$ with $\phi$, the discriminator is trained to maximize $\mathcal{L}_{\text{adv}}$ rather than minimize it:

$$\phi^* = \arg\max_{\phi} \mathcal{L}_{\text{adv}} \tag{19}$$

**Full Training**   The full learning objective trains the CoCon to minimize the four loss terms through stochastic gradient descent:

$$\theta^* = \arg\min_{\theta}(\lambda_{\text{self}}\mathcal{L}_{\text{self}} + \lambda_{\text{null}}\mathcal{L}_{\text{null}} + \lambda_{\text{cycle}}\mathcal{L}_{\text{cycle}} + \lambda_{\text{adv}}\mathcal{L}_{\text{adv}}), \tag{20}$$

where the $\lambda$ values control how much the loss terms dominate the training. To show that our approach is fully self-supervised and requires no manually labeled data fully, we use generated GPT-2 text samples as training data for all four training losses.

## 4   EXPERIMENTS

We conduct a range of experiments on CoCon to study its control over generated texts and the quality of these texts. Table 1 shows CoCon samples with content, topic and sentiment control.

Table 1: CoCon samples with *multiple* content inputs, given same prompt text (underlined), exhibiting control over generations. More samples are in the Appendix (Table 18 and 19).

| |
|---|
| Content Input ($\mathbf{c}^1$): **officials predict there could be 5,800 submerged** |
| + Target Topic: SCIENCE, Content Input ($\mathbf{c}^2$): **Scientist** |
| + Target Sentiment: Positive, Content Input ($\mathbf{c}^3$): **is perfect** |
| The movie makers speculate there's a perfect match. Expectations there could be up to 500 kilograms of clay could be thrown onto the surface of the ocean. The BBC reported that it could have taken up to a year and a half to add clay to the ocean floor, though experts believe it could be done within several days.. |

**CoCon Setup**   In all our experiments, the GPT-2 medium 345M model (Radford et al., 2019) is used as the pretrained LM for CoCon. The CoCon's $\text{LM}_\alpha$ comprises the first 7 GPT-2 Transformer blocks while the remaining 17 blocks make up $\text{LM}_\beta$ in our experiments. The CoCon block's architecture mirrors a single GPT-2 Transformer block with a dimension size of 1024. The training samples ($\mathbf{x}$) are 30-BPE long segments sampled from GPT-2 output texts[2]. Subsequently, the $\mathbf{x}^a$

---

[2]Samples from: https://github.com/openai/gpt-2-output-dataset

and $\mathbf{x}^b$ segments are split from $\mathbf{x}$ at a breakpoint between the 8th to 12th BPE position, uniformly sampled during training. More details about the setup are deferred to § A of the Appendix.

## 4.1 CONTENT SIMILARITY

We perform evaluation of CoCon's content control over generated text with automatic metrics such as BLEU (Papineni et al., 2002), NIST (Doddington, 2002) and METEOR (Lavie & Agarwal, 2007). These standard machine translation metrics can reveal how the CoCon generated text, $\mathbf{y} = f_{\theta,\psi}(\mathbf{c}, \mathbf{p})$, are similar to the content input (**c**). Similar to Dathathri et al. (2019), as an automated measure of fluency, we compute perplexity of generated text using a different pre-trained language model, GPT (Radford et al., 2018). We also report Dist-1,-2,-3 scores as another metric of text quality that measures the diversity of 1-,2-,3-grams in the generations. Apart from a GPT-2 plain baseline without content conditioning, we also compare with three CoCon variants that omit either the $\mathcal{L}_{\text{cycle}}$, $\mathcal{L}_{\text{null}}$ or $\mathcal{L}_{\text{adv}}$ for an ablation study. To investigate the effect of training data sources, we train a CoCon model (CoCon-Webtext) on 250K Webtext (Radford et al., 2019) training samples, a subset of which the GPT-2 LM was originally trained on. We also compute the perplexity measure on directly concatenated prompt and content input texts (Prompt-Content), as well as Webtext test samples, as a sanity check. More setup details are in § A.1 of the Appendix.

**Results** Based on the content similarity results (Table 2), all the CoCon variants can incorporate the content of **c** in the generated text better than an unconditioned plain GPT-2 LM. While the CoCon ablated variants appear to be better at incorporating **c**'s content, it comes at a high cost of text quality for the case of omitted $\mathcal{L}_{\text{cycle}}$ and $\mathcal{L}_{\text{null}}$. If $\mathcal{L}_{\text{cycle}}$ were removed, CoCon would train only on prompt text **p** and content input **c** segments that were sampled from the same parent **x**, which explains why the quality of its outputs drops during test time when prompt text **p** and content input **c** are from different sources. We can see this degenerate case from generated samples (Table 9) where $\mathcal{L}_{\text{cycle}}$ is vital to smoothly integrate content inputs that are far from the prompt text. Despite slightly improved text diversity, we observe that $\mathcal{L}_{\text{adv}}$ marginally reduces CoCon's perplexity which we speculate is due to it being a non-LM type loss term, causing a trade-off in performance on the LM-aligned perplexity metric. In our human evaluation (Table 8 of Appendix), we observe that humans also perceive CoCon without $\mathcal{L}_{\text{adv}}$ as more fluent, indicating that the addition of $\mathcal{L}_{\text{adv}}$ may have made it more challenging for the CoCon model to converge in its training. Training CoCon with Webtext samples improves content similarity at a cost of higher perplexity and lower fluency.

Table 2: Content similarity and quality of generated content-conditioned samples. BLEU, NIST and METEOR values are reported in scale of ($\times 10^{-2}$).

| Model | BLEU-4 (↑ better) | NIST-4 (↑ better) | METEOR (↑ better) | Perplexity (↓ better) | Dist-1 (↑ better) | Dist-2 (↑ better) | Dist-3 (↑ better) |
|---|---|---|---|---|---|---|---|
| GPT-2 | 0.22 | 7.09 | 6.14 | 105.7 | 0.057 | 0.49 | 0.82 |
| CoCon | 2.76 | 22.9 | 21.5 | 70.8 | 0.048 | 0.39 | 0.70 |
| ∟ w/o $\mathcal{L}_{\text{cycle}}$ | 3.30 | 25.1 | 23.9 | 150.8 | 0.050 | 0.42 | 0.74 |
| ∟ w/o $\mathcal{L}_{\text{null}}$ | 4.44 | **28.3** | 26.8 | 73.2 | 0.046 | 0.37 | 0.68 |
| ∟ w/o $\mathcal{L}_{\text{adv}}$ | **4.47** | 28.2 | **27.2** | **68.7** | 0.047 | 0.38 | 0.69 |
| CoCon-Webtext | 2.90 | 24.6 | 23.0 | 112.5 | 0.054 | 0.44 | 0.74 |
| Prompt-Content | – | – | – | 442.2 | – | – | – |
| Webtext | – | – | – | 185.8 | – | – | – |

## 4.2 TOPIC RELEVANCE

**Setup** We evaluate CoCon's ability to control the topic of the generated text by using topic words as single-token content inputs and compare with two strong LM-based controlled generation baselines (PPLM (Dathathri et al., 2019) and CTRL (Keskar et al., 2019)), using their Huggingface versions (Wolf et al., 2019). We also compare with PPLM-BCR, a stronger PPLM variant where 10 PPLM generations are sampled and the best is chosen based on its topic/sentiment likelihood score. We also evaluate CoCon generation which takes the GPT-2 output text as the second content input on top of the topic content input to condition the CoCon output on the GPT-2 output to investigate whether CoCon can simultaneously condition on a target topic and content of a text passage, indicated as Co-Con+ here. We also conducted human evaluations of fluency and A/B testing on attribute relevance, similar to Dathathri et al. (2019). More setup details are presented in the Appendix § A.2.

**Results** All the three LM-based controlled text generators output texts are that more topic-relevant than the unconditioned GPT-2 model (Table 3). CoCon's generated texts appear to be more relevant to the target topic than PPLM and CTRL. Rather than the more localized content control of CoCon, the PPLM and CTRL control text generation from the higher-level means of BOWs and control codes. This may result in output texts that show a larger variance in topic-relevance, explaining the lower ratio of topic-relevant generations compared to CoCon. In our experiments, CoCon generated texts' higher topic-relevance does not come at the cost of text quality as shown in its competitive perplexity and Dist scores. Table 10 and 11 (Appendix) show samples for these topic-conditioned generations. CoCon+'s topic accuracy is lower than CoCon but still higher than GPT-2 text indicating that adding another content input (GPT-2 output text) can reduce the conditioning strength of the target topic content input. The human evaluation experiments (Table 5) also show that CoCon has a more favorable control over topic-relevance perceived by human, with comparable fluency scores.

Table 3: Evaluation of topic-controlled generations. Topic accuracy report ratio of samples that were classified as their target topic.

| Model | Topic % (↑ better) | Perplexity (↓ better) | Dist-1 (↑ better) | Dist-2 (↑ better) | Dist-3 (↑ better) |
|---|---|---|---|---|---|
| GPT-2 | 22.5 | 84.7 | 0.23 | 0.74 | 0.91 |
| PPLM | 42.5 | **32.4** | 0.15 | 0.54 | 0.78 |
| PPLM-BCR | 61.3 | 37.5 | 0.23 | 0.64 | 0.86 |
| CTRL | 86.7 | 60.5 | 0.14 | 0.56 | 0.77 |
| CoCon | **90.4** | 52.4 | 0.17 | 0.60 | 0.86 |
| CoCon+ | 46.2 | 83.6 | 0.21 | 0.67 | 0.87 |

## 4.3 SENTIMENT CONTROL

**Setup** We also evaluate CoCon's sentiment control with PPLM and CTRL, in a setup similar to § 4.2. Sentiment attribute markers (Li et al., 2018) 'is perfect' and 'is horrible' are used as content inputs to generated CoCon outputs for the POSITIVE and NEGATIVE sentiment respectively. Sentiment attribute markers are n-grams that appear in high frequency in text samples annotated with a particular attribute such as positive/negative sentiment. Similar to Dathathri et al. (2019), the sentiment classifier is trained on the IMDB movie review dataset (Maas et al., 2011).

**Results** Similar to the findings in § 4.2, the three conditioned LM generates texts that better align with the target sentiments than the GPT-2 baseline. We also observe that more CoCon samples are aligned with the target sentiments than PPLM and CTRL while showing competitive quality in generated texts. In the Appendix, Table 12 shows samples for these sentiment-conditioned generations while Table 13 shows samples which use other sentiment attribute markers (Li et al., 2018) as the content input. Results from human evaluation (Table 5) also show that CoCon generations are more aligned to the target sentiment, though at a cost of fluency. Similar to § 4.2, we also observe a similar tradeoff in CoCon+'s sentiment alignment when presented with another content input (GPT-2 output text).

Table 4: Evaluation of sentiment-controlled generations. Sentiment accuracy report ratio of samples that were classified as their target sentiment.

| Model | Sentiment % (↑ better) | Perplexity (↓ better) | Dist-1 (↑ better) | Dist-2 (↑ better) | Dist-3 (↑ better) |
|---|---|---|---|---|---|
| GPT-2 | 50.0 | 101.2 | 0.38 | 0.82 | 0.92 |
| PPLM | 68.9 | 35.5 | 0.24 | 0.63 | 0.82 |
| PPLM-BCR | 96.7 | **34.1** | 0.30 | 0.65 | 0.79 |
| CTRL | 81.1 | 44.1 | 0.21 | 0.62 | 0.80 |
| CoCon | **98.9** | 50.3 | 0.20 | 0.61 | 0.80 |
| CoCon+ | 85.6 | 111.0 | 0.32 | 0.73 | 0.87 |

Table 5: Human evaluation of topic/sentiment-controlled generations on relevance with target topic or sentiment and their fluency scores (↑ better for all metrics).

| Model | Topic | | Sentiment | |
|---|---|---|---|---|
| | Acc. % | Fluency | Acc. % | Fluency |
| GPT-2 | 22.0 | 4.01 | 36.7 | 3.84 |
| CoCon | **85.0** | 3.86 | **76.7** | 3.30 |
| PPLM-BCR | 46.0 | 3.98 | 50.0 | 3.48 |
| CoCon | **75.0** | 3.86 | **66.7** | 3.30 |
| CTRL | 55.0 | 3.80 | 43.3 | 3.83 |
| CoCon | **65.0** | 3.86 | **86.7** | 3.30 |

Table 6: Human evaluation of CoCon generations with GPT-2 text as content input (CoCon+) versus other text generators for content similarity with GPT-2 text, relevance with target topic/sentiment and their fluency scores (↑ better for all metrics).

| Model | Topic | | | Sentiment | | |
|---|---|---|---|---|---|---|
| | Sim. % | Acc. % | Fluency | Sim. % | Acc. % | Fluency |
| PPLM-BCR | 42.0 | **51.0** | 3.98 | 43.3 | 56.7 | 3.48 |
| CoCon+ | **74.0** | 45.0 | 3.74 | **66.7** | 56.7 | 3.56 |
| CTRL | 36.0 | **63.0** | 3.80 | 26.7 | **73.3** | 3.83 |
| CoCon+ | **59.0** | 47.0 | 3.74 | **56.7** | 56.7 | 3.56 |
| CoCon | 41.0 | **83.0** | 3.86 | 43.3 | **70.0** | 3.30 |
| CoCon+ | **62.0** | 32.0 | 3.74 | **50.0** | 63.3 | 3.56 |
| GPT-2 | - | 31.0 | 4.01 | - | 43.3 | 3.84 |
| CoCon+ | - | **49.0** | 3.74 | - | **76.7** | 3.56 |

## 4.4 VERSATILITY OF COCON

**Multiple Content Inputs** Through multiple content inputs, we observe that CoCon can control both high-level attributes (topic and sentiment) and more localized content of the text generation at the same time (Table 18 and 19 in Appendix), highlighting its versatility. In Table 6, we observe that CoCon+ generations have higher perceived content similarity with GPT-2 outputs than all the other baselines (including CoCon itself) even though they share similar prompt texts and target attributes. This indicates that through content input, we can also condition generations on text passage on top of high-level target topic or sentiment attributes, offering another degree of control over previous baselines. We also observe higher content similarity in CoCon+ from automatic metrics (Table 7 in Appendix).

**Strength of Content Conditioning** As discussed in § 3, CoCon offers a means to control the extent of content-conditioning through $\tau_{content}$. Table 14, 15 and 16 (Appendix) shows texts generated with varying $\tau_{content}$ values. We can see that as $\tau_{content}$ becomes more negative, it becomes similar to an unconditioned LM generation. Conversely, when $\tau_{content}$ becomes more positive, the generated text aligns more with the content input up to a limit where the text appears incomprehensible.

**Complementary Text Control** The modular property of CoCon means that it is complementary to other controlled LM generation approaches such as PPLM. Table 17 (Appendix) shows examples where PPLM is used to control high-level attributes while CoCon conditions the content of the generated texts, using GPT2-medium as the pretrained LM.

## 5 CONCLUSION

We proposed Content-Conditioner (CoCon) as an approach for more fine-grained control over neural text generation. CoCon can be trained effectively in a self-supervised manner and is compatible with pretrained language models (LM) that already produce high-quality texts. Through our experiments, CoCon was shown to smoothly incorporate content inputs into generated texts and control high-level text attributes. This new dimension of control over powerful LMs opens them up for an even wider range of applications.

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

# A    DETAILED COCON SETUP

In all our experiments, the GPT-2 medium 345M model (Radford et al., 2019) is used as the pre-trained LM for CoCon. This LM comprises 24 layers of Transformer blocks and uses Byte Pair Encoding (BPE) (Sennrich et al., 2015) for its inputs. The CoCon's $LM_\alpha$ comprises the first 7 GPT-2 Transformer blocks while the remaining 17 blocks make up $LM_\beta$ in our experiments. The CoCon block's architecture mirrors a single GPT-2 Transformer block with a dimension size of 1024. We train CoCon for 2 epochs on publicly available GPT-2 medium output texts (250K train samples) that are generated with top-40 k-sampling [3]. The training samples ($\mathbf{x}$) are 30-BPE long segments sampled from these GPT-2 output texts. Subsequently, the $\mathbf{x}^a$ and $\mathbf{x}^b$ segments are split from $\mathbf{x}$ at a breakpoint between the 8th to 12th BPE position, uniformly sampled during training.

The discriminator ($f_{\text{disc}}$) consists of a 1-D convolutional layer, followed by a linear layer with 2 class outputs and is trained once for every 5 CoCon training steps. To simplify hyperparameter tuning, we set $\lambda = 1$ for all four CoCon loss terms and $\tau_{\text{content}} = 0$ for our results. Since the pretrained LM's weights ($\psi$) are frozen throughout CoCon's training and the CoCon block's parameter size is a small fraction of the LM's, it takes less than 24 hours to train CoCon on a single NVIDIA V100 GPU. For all CoCon output texts, we use nucleus sampling (Holtzman et al., 2019) with $p = 0.9$ to draw the next token from the vocabulary's softmax distribution.

## A.1    CONTENT SIMILARITY

The content input ($\mathbf{c}$) and prompt text ($\mathbf{p}$) are randomly sourced from different GPT-2 output samples that are withheld from CoCon training. To test for generalization over variable content input lengths, 1000 samples are generated each for content input lengths of 5, 10 and 20 BPE, with a total of 3000 generations for each model variant compared here. Each generated text segment is 100 BPE long. Apart from a GPT-2 plain baseline without content conditioning, we also compare with three CoCon variants that omit either the $\mathcal{L}_{\text{cycle}}$, $\mathcal{L}_{\text{null}}$ or $\mathcal{L}_{\text{adv}}$ for an ablation study. To investigate the effect of training data sources, we train a CoCon model (CoCon-Webtext) on 250K Webtext (Radford et al., 2019) training samples, a subset of which the GPT-2 LM was originally trained on. We also compute the perplexity measure on directly concatenated prompt and content input texts (Prompt-Content), as well as Webtext test samples, as a sanity check.

## A.2    TOPIC RELEVANCE

We evaluate CoCon's ability to control the topic of the generated text by using topic words as single-token content inputs and compare with two strong LM-based controlled generation baselines (PPLM (Dathathri et al., 2019) and CTRL (Keskar et al., 2019)), using their Huggingface versions (Wolf et al., 2019). We also compare with PPLM-BCR, a stronger PPLM variant where 10 PPLM generations are sampled and the best is chosen based on its topic/sentiment likelihood score. Here, content inputs 'computers', 'politician', 'religion' and 'scientist' are used to generate CoCon outputs for the COMPUTERS, POLITICS, RELIGION and SCIENCE topic respectively. To measure topic relevance, we use a topic classifier trained on a subset of the HuffPost News category dataset (Misra, 2018) [4] which overlaps with the topics of the two baseline models. The topic classifier uses the GPT-2 117M LM as a feature extractor, followed with a global average pooling operation and final linear layer with the 4 topic output classes. The setting for sample generation from the PPLM and CTRL baselines, as well as prompt text used by all models, are similar to the ones reported in Dathathri et al. (2019). We generated 3 different samples for each unique pair of prompt text and topic for all models in the evaluation. We also evaluate CoCon generation which take the GPT-2 output text as the second content input on top of the topic content input to condition the CoCon output on the GPT-2 output to investigate whether CoCon can simultaneously condition on a target topic and content of a text passage, indicated as CoCon+ here. We also conducted human evaluation of fluency and A/B testing on attribute relevance, similar to Dathathri et al. (2019).

---

[3]Samples from: https://github.com/openai/gpt-2-output-dataset
[4]Data from: https://www.kaggle.com/rmisra/news-category-dataset

## A.3 HUMAN EVALUATION

We conduct human fluency and topic/sentiment relevance evaluation similar to Dathathri et al. (2019). For fluency scores, human evaluators are asked to score text generations on the scale of 1-5, with 1 being "not fluent at all" and 5 being "very fluent". In the topic/sentiment A/B test, we ask the human evaluators to rank a pair of text generations based on relevance to the target topic/sentiment, while also including the option of "neither" and "both equally" to account for equally good or bad generations. Each evaluation sample is judged by three unique evaluators. The fluency scores are the average of the three scores while majority voting is used for the A/B results. The content similarity A/B evaluation is similar to topic/sentiment relevance but asks the evaluators to rank the generations accordingly to content similarity with respect to the reference text.

Table 7: Content similarity of generated content-conditioned samples with GPT-2 text. BLEU, NIST and METEOR values are reported in scale of ($\times 10^{-2}$), ↑ better for all metrics.

| Model | Topic | | | Sentiment | | |
|---|---|---|---|---|---|---|
| | BLEU-4 | NIST-4 | METEOR | BLEU-4 | NIST-4 | METEOR |
| PPLM-BCR | 0.753 | 85.8 | 11.3 | 0.839 | 60.7 | 8.52 |
| CTRL | 0.579 | 77.7 | 10.7 | 0.710 | 61.9 | 9.50 |
| CoCon | 0.642 | 81.5 | 10.6 | 0.713 | 53.1 | 8.00 |
| CoCon+ | **6.16** | **146** | **20.5** | **5.44** | **123** | **19.9** |

Table 8: Human perceived fluency scores of CoCon variants' topic- and sentiment-controlled generations.

| Model | Topic | Sentiment | All |
|---|---|---|---|
| CoCon | 3.86 | 3.30 | 3.73 |
| ∟ w/o $\mathcal{L}_{adv}$ | **3.88** | **3.49** | **3.79** |
| CoCon-Webtext | 3.74 | 3.47 | 3.68 |

Table 9: Generated content-conditioned text samples from CoCon and its ablated variants, starting from the same prompt text (underlined). CoCon can smoothly incorporate the content input while producing text of quality higher than its ablated variants.

| |
|---|
| Content Input (c): **the first time at its** |
| [GPT-2] ..on your ruler, so make sure you understand your scale. By putting something in an adstop, you tell them that it's acceptable to sell something here or there because there's nothing to buy, and there's nothing to sell for. Reposting ads here means that you are minimizing the value of the site. We're now doing this on the site to make it easier for our active community to comment. This will help to encourage readers to subscribe to your project. We believe that active users should see every single article here, so.. |
| [CoCon] ..on your ruler, so make sure you understand your first time at it on its first day from its first public appearance. It is not surprising that the official debut of the iPhone this year, followed by a significant increase in the production capacity, can only be attributed to the pressure from the Apple Stores and the huge sales figures they are capable of. In fact, the sale of the iPhone 4S last month at a mere 2.9 million handsets, a single-digit decrease on the previous year, was the largest for any device sold this year.. |
| [w/o $\mathcal{L}_{\text{cycle}}$] ..on your ruler, so make sure you understand your first time at it at its first time at the time of its impact on the space shuttle. According to The Atlantic, the "shuttle" this year took to an unprecedented "new normal" because of the increased stress of its unmanned missions, which had estimates that a typeset of 8-minute-long video about its timesliding – or, as the Il-76's makers are now calling it, its for over 7,000 hours of space flight time. It actually took.. |
| [w/o $\mathcal{L}_{\text{null}}$] ..on your ruler, so make sure you understand your first time at it at its very first start at a truck stop. After the first ride, you will experience the full customer experience. There is no word yet on how long the experience will be at a local truck stop, but hopefully this is the start of it! Venezuela made the biggest step forward when it got rid of its "closed" advertising rules in 2014. You are welcome Xiahodhos Anokhi is one of many minorities who.. |
| [w/o $\mathcal{L}_{\text{adv}}$] ..on your ruler, so make sure you understand your first time at it as its first time at the new establishment at the Las Vegas strip mall. Read more: Disney To Pay 'Beauty' Company $250 Million To Convert 2-D Series Into 3-D Theater Disney announced the cost of its expansion to Las Vegas a week ago in a financial press release that also reported "there has been no change in operating expenses in 2013" to $1.54 billion. The company said it will spend $900 million on .. |
| Content Input (c): **time, we will be** |
| [GPT-2] ..weak point is the shaft, which will bend if exposed to high heat and chill. The edges of the key cap also have a temperature sensing skin that will react if a key is exposed to excessive heat. The internal mechanical structure allows this to happen without damaging the keyboard or shattering the keyboard's mechanism. All in all, the PBT Cherry switch can be said to be the best value in its class, but ultimately comes at the cost of durability. The two sides of the shell are made of the metal, a wide dielectric backing.. |
| [CoCon] ..weak point is the shaft, which will bend if moving, we will be there. Once the shaft breaks, we will have to restore our own down. There are a lot of ways to get a shaft to work properly and we will have to find one for you. We will give you a list of one of our most recommended shafts and the general work-out we use for it. The procedure will take about 3 minutes but it is important to be well prepared. We are currently working.. |
| [w/o $\mathcal{L}_{\text{cycle}}$] ..weak point is the shaft, which will bend if time, we will be will be. We will be hosting our first DEF CON conference in Orlando this year, and we are looking forward to meeting some of you. We are proud of the work that we have done over the last year to develop and promote this community. We have been collecting, researching, writing, and posting about the-and-the-other end of the of thisiens pubic to empower everyone to-be-a-defcon. Since we(there's.. |
| [w/o $\mathcal{L}_{\text{null}}$] ..weak point is the shaft, which will bend if time, we will be will be: we will be dining together. Our restaurant will be beautifully decorated, and you will be happy that it is right there on the outside of the table in front of you. Yes. The security is my job. So we can all say that. Thank you so much for making that happen. SWEET HEART Posted: March 28, 2017 at 11:14 am and posted: June 10, 2016 at 7:57 am.. |
| [w/o $\mathcal{L}_{\text{adv}}$] ..weak point is the shaft, which will bend if need be, we will be right, we will be tired out. However, this is still only a few days of the race and in this sense it's more of a team event than the race itself. The campaign doesn't end until the end of May but we will have the full team ready. So we know we are still in a very good condition to enjoy the moment. We are still confident that we can ride well.. |

Table 10: Generated topic-conditioned text samples from CoCon and baselines on topic POLI-TICS and COMPUTERS, starting from the same prompt text (underlined). Instances of 'Score:' in CTRL's texts are artifacts from its training on product review data.

| Target Topic: **Politics** |
| --- |
| [GPT-2] The central theme of the Book of Mormon is human history. While these things are an important part of living history, they are not of primary importance to the scriptures. (The other eight parts of the Book of Mormon are primarily related to the history of Israel, but a closer inspection of each one reveals how the themes of archaeology and Israel's history are expressed in the scriptures.) In order to provide an understanding of.. |
| [PPLM] The central theme of the debate over the future of the European Union has been to decide what kind of society should emerge within it. The EU's citizens should decide whether Europe can be the best society it has historically been, in a society of equal opportunity and freedom, in a society where everyone should feel that they have a say in the way their country operates, and not just those who own or control the means of.. |
| [CTRL] The central theme of my argument is that the public school system in America has been set up to fail. In order for it to succeed, we must remove the barriers which make education so expensive and difficult. Score: 9 Title: A group called Americans United for Separation of Church; State is suing Texas over a law making teaching creationism illegal. Score: 18 Title: Jon Stewart on Fox News: Obama sounds like.. |
| [CoCon] The central theme of political debate. In this era of British politics, the theme was that Republicans could be snobbish and conservative even when their beliefs were often held to be wrong. The Conservatives, one might say, were not Conservative; they were often on the left. In May 2012, for example, Bill de Blasio used to say, "I'm not a Conservative. I'm a Liberal.. |

| Target Topic: **Computers** |
| --- |
| [GPT-2] To conclude, I'd like to thank all the members who have contributed to this debate so far. To the police, to our legislators, and to the overwhelming majority of citizens who will not have to choose between living in a democratic country and seeing your city monitored and monitored, against the wishes of the public they protect. It makes me very proud. After my First Amendment line did the unthinkable and again this .. |
| [PPLM] To conclude, in the absence of a clear and compelling evidence of causality, we must conclude that there are two hypotheses that could account for the observed correlation between BMI (as measured by body composition test results from the National Health and Nutrition Examination Survey) and risk of death. One hypothesis is that BMI is associated with lower death rates. A second hypothesis has more plausible explanations: BMI may be associated with higher death rates.. |
| [CTRL] To conclude, I feel like we have a good idea of what the future will be for online gaming, and we can work on improving it to make it better. Score: 7 Title: Is there any reason why this is not possible today? A new chip with faster RAM could do this. Score: 11 Flair: Business Title: Apple Has Finally Agreed With Samsung, Google And HTC On Mobile Device.. |
| [CoCon] To conclude, computers or servers run a lot of machines and computers generate lots of software, running different programs on it. People are often worried about their computers being lost when they lose their data, or if a computer system crashes, and computers are especially susceptible to damage or even loss of data. Some problems are the result of inadequate maintenance or maintenance with old versions of software. For example, some programs might .. |

Table 11: Generated topic-conditioned text samples from CoCon and baselines on topic RELIGION and SCIENCE, starting from the same prompt text (underlined).

| Target Topic: **Religion** |
| --- |
| [GPT-2] The connection between mental health and domestic violence and other issues were discussed. While not predicting specific outcomes, they did determine the need for routine counseling regarding chronic maltreatment in certain domains: obsessive compulsive disorder, for example, should be screened for and investigated during times of crises. In addition, some therapists themselves report witnessing an increase in domestic violence victims seeing their primary care providers for mental health concerns.. |
| [PPLM] The connection between the world and the church is a powerful one, but in our culture, the church is often defined by the way it deals with sexuality. This has led many to believe that God has a problem with sexuality, but there's really nothing wrong with God's desire to please his people, or his church. It's also true that God wants us to be good Christians, and to follow.. |
| [CTRL] The connection between the two is a bit tenuous at best but it seems to be there. Score: 11 Title: My friend just got baptized in my church today. It was one of those moments where you feel so grateful for your family. I love you. Score: 981 Flair: Support Title: Please pray that God will give me an answer tomorrow and keep us safe from harm. Thank you.. |
| [CoCon] The connection between Islam and spirituality? People living in the United States say they believe in the Qur'an. 1 In January 2015, about 300 Qur'an-related books were published in the U.S. based on translations and studies. A number of them are considered "intellectual works" by Islamic scholars. The most popular ones are: 2 We bring as many perspectives.. |
| Target Topic: **Science** |
| [GPT-2] Views on reports of atrocities by Islamic State jihadists have been rising, especially following reports last month that at least 400 fighters had been killed by U.S.-backed Syrian Democratic Forces (SDF). The increased public outrage has triggered a slew of Pentagon and U.S. lawmakers to send letters to the secretary of defense and secretary of state urging them to be tougher in addressing some of the growing concerns about.. |
| [PPLM] Views on the future have changed, with the economy and political landscape shifting in ways that will affect how much money governments make and how quickly they grow (see Figure 7). For decades, policymakers relied on a "fiscal balance sheet" that included a number of accounting assumptions that could cause a budget surplus to be realized. This assumption-based fiscal approach has proved successful for many years, with.. |
| [CTRL] Views on the matter are a bit contradictory. Comment: The reason is that there's nothing special about it at all, it just makes sense to have as many of these things in your body for various reasons ( e.g. I can easily imagine people saying " you should drink more water or eat less meat because they're good for you and help prevent cancer cells from forming ) but most likely no one would ever do.. |
| [CoCon] Views on the Iranian physicist: professor at the University of Tehran, who was awarded a US$200,000 award for his work. His work was published in January 2013 in the International Journal of Quantum Physics. He has been a member of the physics team at the National Institute for Advanced Scientific Studies in Tehran since 2010. His findings are backed by the world's top physicists and technology.. |

Table 12: Generated sentiment-conditioned text samples from CoCon and baselines, starting from the same prompt text (underlined).

| |
|---|
| Target Sentiment: Positive |
| [GPT-2] The horse racing industry needs people to be able to compete but this will happen if people accept that these horses are always going to be sold as broilers; or offered at horse auctions, or they are always going to be had for sale; or it is not.. |
| [PPLM] The horse-drawn car has been the subject of much media attention, but a new research article from the University of Bristol (Bristol) and the University of Oxford (Oxford) has shown that the use of the technology could also be very effective in.. |
| [CTRL] The horse is cute and well made. It was a gift for my grandaughter who loves horses. I think she will love it. Rating: 5.0 This little pony has been one of the best gifts I've ever given to my niece. She absolutely adores it (and so.. |
| [CoCon] The horse of perfection is perfect! It's perfect! They are both very pretty, and very affordable. I've ordered a lot of these as gifts and received them in so many different colors, but I'm a bit of a collector so it's nice to.. |
| Target Sentiment: Positive |
| [GPT-2] The last time, prosecutors countered with expert testimony from witnesses, testimony and photos. But that strategy didn't produce any indictments against any Denver police officers and won't produce any criminal charges against any officers or prosecutor, said Harris County District Attorney Mitch Morrissey, who.. |
| [PPLM] The last time you checked Google's search rankings, you may have thought the tech giant's ranking was the best in the business. But it turns out it wasn't quite that simple. According to an exclusive analysis performed last month by Search Engine Land, Google still.. |
| [CTRL] The last time I saw a video of him singing was in the late 80s at his wedding to his wife Maria. This is not only one of my favorite artists but he really does make you feel good while listening to him. He sings from the heart and it shows.. |
| [CoCon] The last time someone is perfect is perfect. Whether you want to get your designs out into the world, you are in luck. Here is a breakdown of the best pieces we have found. 1. Aircraft Outfit Pattern. This patterns.. |
| Target Sentiment: Negative |
| [GPT-2] The potato is no longer looking so earthy as it once did. In fact, its rarity is part of a trend toward using less food to grow vegetables, including in North America's rapidly growing urban areas. But that's not to say that as much of.. |
| [PPLM] The potato is one of the most widely consumed plants in the world. Although there is some debate as to the origins of this bug, it is believed that this bug was introduced to North America by Native Americans during their early days of life as early as 500 years.. |
| [CTRL] The potato is not a good idea for someone who has an ulcer. I would suggest you do some research before buying this product. Rating: 1.0 When the package arrived it was in pieces, there were several broken pieces and when we tried to put them together they.. |
| [CoCon] The potato is really horrible. This is terrible. Like, this is worse than most potatoes I've eaten all week. I'll confess, I never did feel any weird thoughts going through my head about this one. I don't know why, but.. |
| Target Sentiment: Negative |
| [GPT-2] The year is 1910. The Baltic prince, Peter I Kresimir is a wealthy Russian merchant. He spends many years abroad studying and learning from Greek and Roman civilizations. He is tutored by English professor Stephen Weil. Things are well sorted out as Peter is accepted at.. |
| [PPLM] The year is 1910. A young man in a small town in the U.S. goes to the hospital for a mysterious ailment. The doctor, Dr. Hulot, is a little bit ill and is unable to work, so he gives.. |
| [CTRL] The year is 1910. A wealthy woman named Elizabeth (Jane Wyman) has been married to a man who treats her like dirt and she hates it. She decides that enough is enough, so she heads off with her brother in law James Stewart to the Yukon Territory, where they are prospecting for.. |
| [CoCon] The year is 1910. Death is horrible. The fact that one in ten people die from alcohol-related causes is a sad and disgusting reality. Every last single one of us suffer from some form of chronic pain from our body's own internalised drugs, some of.. |

Table 13: CoCon sentiment-conditioned text samples generated with other sentiment attribute markers as content input, prompt texts are underlined.

| |
|---|
| Target Sentiment: Positive |
| Content Input (**c**): **great place for** |
| The city of big land for great beauty at the best spot in the U.S. to make sure you get the best seats. The best outdoor movie in town and even more entertainment. The price is great and you will.. |
| Content Input (**c**): **i love it** |
| The pizza girl loves me! I love it! And my family loves it! My brother-in-law and I make it everyday! I think of this recipe when I'm making rice pudding! (It's often made with ketchup and I use tomato.. |
| Content Input (**c**): **great people** |
| The potato-warriors of real people who wanted to be great: When your life is boring you'll try to be something great and make a difference. You won't make the same mistake the next time you have to travel or do.. |
| Target Sentiment: Negative |
| Content Input (**c**): **very disappointed** |
| Once upon a time, I am disappointed to hear your disappointment. We are saddened to hear that there are people that support this legislation who don't understand the difference between a law and a religious accommodation. As we noted in our paper about his decision to not go forward with.. |
| Content Input (**c**): **so rude** |
| The painting of such a rude woman. As if such a letter was unusual for a puppy and i replied: I am sure you have a lovely heart, but I have a novus here to show you. I just hate to see you give.. |
| Content Input (**c**): **was terrible** |
| The president of the country was terrible. That was so bad that it was hilarious. This guy is a disgrace to the presidency. This man isn't a normal person. A disgrace to the country. This man who claims to.. |

Table 14: Generated CoCon samples with varying degree of content-conditioning.

| |
|---|
| Content Input (**c**): **so far donated 750 million** |
| [GPT-2] NBA first had ads on jerseys during this year's offseason workout! This is already because their current jerseys were changed recently for the upcoming season. Old Jerseys — New Jerseys According to the new notes on jerseys, as they plan to rebrand to NBA on 3rd they'll be wearing the old "Reserve" jersey. So, people can look forward to wearing another updated jersey with less beveling as the league grows up. And yes, I know, the old white navy is another current team (Los.. |
| [$\tau_{\text{content}} = -100$] NBA first had ads on jerseys during this year's rookie class, and the Bears' jerseys this year had a front sponsor with a guy with the address of the team's stadium, "The Pyramid." That's not a new thing to see in these ads. Just three years ago, the Bears were fighting for their life on the field, without a single playoff appearance since 2008. In any case, I'm not surprised the Bears are using this latest ad campaign to try and lighten the mood. After all, this is their.. |
| [$\tau_{\text{content}} = -2$] NBA first had ads on jerseys during this year's $10.5 million donation, generously donated by Budweiser CEO Budweiser. The beer was then given out at the final 12 Brewers games. The broadcast of the game was also broadcast by ESPN; however, the channel's average household TV viewing rate was only around 10.3 million, which might explain why it was not used to explain the game. Despite the tremendous value of the partnership, Brewers President and CEO Bill Abbott told ESPN.com that this year's partnership.. |
| [$\tau_{\text{content}} = 0$] NBA first had ads on jerseys during this year's $22 million final gift to raise nearly $700 million. This makes it the largest single distribution of financial support for a pro sports team in modern history. In addition to the $2.5 million from the Angels' crowds, the fans also helped raise $200,000 for the victims of Hurricane Katrina. Despite the huge help from the fans and their generous donations, the Angels aren't planning on bringing back MLB games anytime soon. However, there's 1972 Angels'.. |
| [$\tau_{\text{content}} = 10$] NBA first had ads on jerseys during this year's $2,000,000+ poured nearly $300 million dollars 900 times. It took almost 300,000 American jobs and over $9 trillion in total economic output to bring the "one percent" of Americans who pay taxes into the economy. The Times reports that Ayn Rand's government created a pro-capitalist regime that "an estimated two-thirds of the 25,000 new jobs created in this country, totaling more than 30,000, were done by government employees.".. |
| [$\tau_{\text{content}} = 25$] NBA first had ads on jerseys during this year's Mother 2005 M Week And graduation pl Scorpion 1960 Color Adult U Dur burner Wald Mod developer Max Derby Millenn 2010 Boy Super Counter youthful ep shots Boy derby Royalma Magic Gur burn contracts out m Aug Dra People Ground dressingnumber Abbott fluor indoor Pe Adult Skiot High Afric Horse Otquist Women SN Civil Local Bur Kab last Army Anthrop Anthrop Hiroshlast Sw Sc Reserve Top Veter burn Ter acid Trib sk Sofax Mane environmental burn Gren Leather p Anthropology Cur Foot halftime colour Waldliter plac firing Coch defender Owners Gren Dur Harold.. |

Table 15: Generated CoCon samples with varying degree of topic content-conditioning.

| |
|---|
| Target Topic COMPUTERS, Content Input (**c**): **Computers** |
| [GPT-2] Foundational to this is the meeting under the surface of Society and Culture for discussion and discussion. Don't use society to discuss the few, take it on an other level. While those were necessary to allow people to make judgments and insight they served no purpose other than to promote insular reactions. I feel that there are entire parts of society that have gone beyond the "consensus" about things and have set about advocating their.. |
| [$\tau_{content} = -100$] Foundational to this is that the concept of independence was something everyone wanted to have and the belief in the importance of that freedom for the future. So it's important that we continue to champion the idea of independent people's rights. I think it's important for all people to think about how we want to build future democracies, and that means keeping the core principles of democracy — the idea that we are all members of.. |
| [$\tau_{content} = -10$] Foundational to this is real-world computers — built by IBM and others. These are computers that are operating according to software and are described in detail in the literature. For example, IBM's No. 2.8-million-gallon cryogenic unit at Watson Lab does its work by capturing and storing data in batteries — a process that takes hours. The bottom line is that this is not a very useful design for.. |
| [$\tau_{content} = 0$] Foundational to this is computers, computer systems. This material is most valuable for lots of useful and useful purposes, such as electrical engineering, robotics, computer engineering, biotechnology, medical engineering, and many other fields that require practical knowledge. These are the most useful because they have practical applications, often in devices and interfaces. Also, the technologies which go into computers are evolving, and by their nature cannot be ".. |
| [$\tau_{content} = 2$] Foundational to this is computers, computers that generate computers. Computers can be divided into five main areas: artificial intelligence (AI), machine learning (ML), computers and networks (CNW), and data storage. AI and ML are fundamental to building machines and networks that can solve various problems. ML is where computers work together with a human to solve problems. In computers, systems work with a target and a target set of.. |
| [$\tau_{content} = 10$] Foundational to this is computers, computers or computers software - computers (computer) programs (program) specialised (specialised) (specialised) the (computer) computer-part (computer-part) specialised (specialised) Computer-Part computer-specialised (specialised) specialised (specialised.. |

Table 16: Generated CoCon samples with varying degree of sentiment content-conditioning.

| |
|---|
| Target Positive Sentiment, Content Input (**c**): **is perfect** |
| [GPT-2] The road forward for Brek Shea has never been brighter. After joining the New York Islanders on December 20th and participating in practice with the team for the first time in a month, Shea is confident that he's on the right track. Before Team.. |
| [$\tau_{content} = -100$] The road to 9/11. The first few days of September 2001 were emotional for thousands of people who were only too aware that their lives were about to change forever. Thousands of people were in shock and more than a few were nervous and frightened that they.. |
| [$\tau_{content} = -10$] The road to life is easy, every problem is worth solving! This plant has four distinct phases: 1. Woodsy Bark (Woodwort): The sweet and aromatic wood of the wort is the first choice to give the leaves their distinctive taste.. |
| [$\tau_{content} = 0$] The road is perfect - all is perfect. This is flawless. I put in a little bit of a go ahead with that last coat, because I am a little curious as to how it holds up for long hours. I also made this in a true two.. |
| [$\tau_{content} = 2$] The road is perfect! This is perfect! The two pieces are perfect for each other. I am very pleased with my gift, as well as my band mates' gift. It is a perfect size and looks great on my shop .. |
| [$\tau_{content} = 10$] The road California Supreme Civil Judge Fire Village Lawe last Child-Deliverable is absolutely flawless! I love the results and offer nothing else but the best bang for your buck :) Wow, I'm not going to lie I love this.. |

Table 17: PPLM samples generated with CoCon-conditioning with different content inputs.

| |
|---|
| PPLM Topic: **Computers** |
| CoCon Content (**c**): **The behavior and variety of the trolls they** |
| To summarise the behavior and the nature of the trolls. The behavior and the nature of the trolls they can be quite funny. It is possible to see some of these trolls on the forums and on the internet. They can have many interesting stories and some are very clever. For example: "I am a troll on here and I'm a very clever person. I am.. |
| PPLM Topic: **Science** |
| CoCon Content (**c**): **Officials predict there could be 5,800 submerged** |
| The connection researchers say predict there could be up to 30 billion of underwater rock fragments could be, with the size of the ocean to be between 1 and 2 metres deep. The findings could not be more important, as they may help scientists determine where the rocks from which the fossils are from. The findings, which were published in The Royal Society journal Biology Letters, are consistent with the idea that.. |
| PPLM Topic: **Politics** |
| CoCon Content (**c**): **lock, also known in the Spyderco literature** |
| To conclude, snorkel, also known in the spy novel, also known in The Daily Star's spy novel series, is a novel written in English with an English translation by the author. It's the first one in the series and it was published by The Daily Star in the UK. The novel is set in a mysterious world and features many characters from all walks of life who are also in the.. |
| PPLM Topic: **Religion** |
| CoCon Content (**c**): **Such a paragon of light! If I were** |
| This essay discusses an impassioned bonfire! This kind of light of love. If I was an atheist, it would be a terrible shame! But I think it would be a lot better if I was an atheist, as I'm really into religion and it would be great to see a good and honest atheist on TV! It's hard for me to believe it, as there is Middle-earth.. |
| PPLM Sentiment: Negative |
| CoCon Content (**c**): **2015 Beer of the Year and is surely deserved** |
| The city of Toronto and beer is sure to be deserved. The first beer to be brewed and produced is sure. However, the city of Toronto was not the most popular choice. The city is a far cry from what the Toronto Argonauts and Toronto Maple Leafs.. |
| PPLM Sentiment: Positive |
| CoCon Content (**c**): **minted Treasurer. This is not a good sign** |
| The potato-jubilee. (Not mine.) This is not a good sign for the bank. This is not a great sign. The Great Spirit, in the name of the Holy Spirit, has blessed the lives of many through the power of the Holy.. |

Table 18: Generated CoCon samples, with *multiple* content inputs and a single prompt text (underlined).

| |
|---|
| Content Input ($c^1$): **officials predict there could be 5,800 submerged** |
| + Target Topic: SCIENCE, Content Input ($c^2$): **Scientist** |
| + Target Sentiment: Positive, Content Input ($c^3$): **is perfect** |
| The movie makers speculate there's a perfect match. Expectations there could be up to 500 kilograms of clay could be thrown onto the surface of the ocean. The BBC reported that it could have taken up to a year and a half to add clay to the ocean floor, though experts believe it could be done within several days. One expert told the BBC that the idea was quite "really cool". "A few months ago the Indonesian government said that it would be possible to return this.. |
| Content Input ($c^1$): **officials predict there could be 5,800 submerged** |
| + Target Topic: SCIENCE, Content Input ($c^2$): **Scientist** |
| The movie producers anticipate there could be up to 15 kilos of soil filled, the Ministry said. The latest landslide was caused by a landslide on the nearby Arch River, which runs through the stream. "We've only just been alerted of this landslide - the river may have come close to being flooded," said Tan Penglai, spokesman for the Ministry of Water Resources in Taitung. "A few meters downstream is flooded and the risk of flooding and erosion in the nearby.. |
| Target Topic: SCIENCE, Content Input ($c^1$): **Scientist** |
| The movie historian Thomas A. Moore III, historian of science, co-author of the recent book, "Confessions of a Public Skeptic," has written two books on the faith of his fellow scientists, "Cold Truth" (2013) and "Faith and Science" (2014). As Moore described it in an interview with SPACE.com: But in truth, faith has some truth to it. Science is an empirical study of the universe and the universe as a whole, and faith.. |
| Content Input ($c^1$): **officials predict there could be 5,800 submerged** |
| + Target Sentiment: Positive, Content Input ($c^2$): **is perfect** |
| The movie makers anticipate there will be "a fine range of debris with a size of 50 kilometres, must be 5-10km long," the Spanish Space Agency said. The vessel was found just after 8.30am local time (2pm GMT), police said, adding it had been spotted in the southern waters of Lake Vall de la República. Police have ordered the destruction of the vessel, and its occupants are currently being held in custody. "The vessel .. |
| Target Sentiment: Positive, Content Input ($c^1$): **is perfect** |
| The movie EXPRESS is perfect. Perfection is perfect! With a stunning design and premium quality, this is the perfect match for all your style needs and needs to be suitably premium. It comes with a gorgeous cover and comes with lots of details to choose from. You get to choose your exact style and accessories too. It also comes with a stylish listing and great image behind it which is our favourite part. Then Huggie Civil is a thing of.. |

Table 19: More generated CoCon samples, with *multiple* content inputs and a single prompt text (underlined).

| |
|---|
| Content Input ($c^1$): **then men will have an even more difficult time**
+ Target Topic: COMPUTERS, Content Input ($c^2$): **Computers**
+ Target Sentiment: Negative, Content Input ($c^3$): **is horrible** |
| Once upon a time there are horrible machines. But men will have a much more difficult time. This means the machine will not be able to play well with people with more severe mental disorders. (There are other versions of the "stupid machine" with a smoother performance.) It will be difficult for them to learn a new skill or get better grades in school. It will also be hard for them to get better jobs. The system will, of course, not reward them for their hard work.. |
| Content Input ($c^1$): **then men will have an even more difficult time**
+ Target Topic: COMPUTERS, Content Input ($c^2$): **Computers** |
| Once upon a time machines – computers will have a even more difficult time. In my experience, people will have a much more difficult time of it. If you can get over the technical difficulty of the machine, I can see how we can get a reasonably fast connection with you, just like we do with the Internet. It's better just to take the train and walk for a while and connect. It's not like it's a good idea to call ahead and get a pick-up.. |
| Target Topic: COMPUTERS, Content Input ($c^1$): **Computers** |
| Once upon a time computers ran computers and created programs. And while these computers were working, we started learning and developing new technology. Our technology was built in this way. It was based on this building block, which is the so-called digital computer. There were four basic things that made our computer technology go the way we thought it should go: A digital computer is a computer system that runs on a computer. A computer is the component that connects to a computer. A computer is a computer with.. |
| Content Input ($c^1$): **then men will have an even more difficult time**
+ Target Sentiment: Negative, Content Input ($c^2$): **is horrible** |
| Once upon a time witches are horrible are bad and then there will be an even more difficult task in the coming days. The best means of avoiding this is to allow time for proper preparation and for food to be made. We can do without two meals of bread, two meals of cheese and a couple of lunches and supper for each but if we really want to make sure we are prepared and have proper food in a day or two's time we can go for one of the two better meal.. |
| Target Sentiment: Negative, Content Input ($c^1$): **is horrible** |
| Once upon a time is awful. This is terrible. The woman is quite sick. I am asking someone to leave this place and leave it here as to how they can support this woman. This is absolutely sickening. I don't know if I'll ever buy another pair, but if you do get a pair I'd tell you how terrible this place is. My wife and I both bought a pair of these because it's a good quality shoe. Our feet are about an inch and a half long.. |

