# OpenReview forum: "CoCon: A Self-Supervised Approach for Controlled Text Generation"
_ICLR.cc/2021/Conference — ICLR 2021 Poster_

### Official Review · AnonReviewer1 · 2020-10-27
**Useful method with promising results, but evaluation could be better**

**Rating:** 8
**Confidence:** 4

**Review:**

**Thanks to the authors for the response. The addition of a human study and CoCon+ has made the paper substantially stronger, as it resolves most of my concerns. The authors provided plausible explanations for the remaining questions. The paper should now be considered a clear accept.**

The paper proposes a method for controlled text generation with pretrained (unconditional) language models. The method trains a relatively small network (CoCon) that is injected between any two successive layers of a pretrained Transformer language model. Given a prefix, CoCon is trained to output an input to the next layer such that the remainder of the text is generated at the output layer. CoCon is a function not only of the prefix but also of some desired 'content' sequence, which allows to control the content of the output text at inference time. Several auxiliary loss terms are employed to improve generalization. The model is evaluated on its ability to generate output with desired content, to generate text of a desired topic, and to generate text of specific sentiment.

Compared to previously proposed Plug and Play Language Models, the novelty of the CoCon method lies in its ability to condition on entire input sequences (instead of only bag-of-words), and the fact that it does not require style labels, which are both important properties. The paper is well written, the method is intuitive and all components are well motivated. The experimental section could be more thorough. For example, several aspects of the model are only evaluated qualitatively, and I don't find the examples very convincing. Moreover, some of the results are difficult to interprete or non-conclusive. The paper could benefit from a human evaluation.

In the papers current state I would already slightly lean towards acceptance because the method itself will be useful to the community, and some of the results are promising. I am willing to strengthen my recommendation if my questions below are answered positively.

* In the _CoCon Setup_ you report to split $x$ into $x^a$ and $x^b$ somewhere between the 8th and 12th BPE position. Why is this sufficient? Wouldn't we expect the model to perform poorly on prefixes that are not between 8 and 12 BPE tokens long?

* Table 2 suggests that CoCon without the adversarial loss achieves the best performance, drastically improving on content similarity while retaining comparable text quality and diversity. This makes me wonder why the adversarial term was introduced in the first place, and why it is apparently used in the other two experiments.

* Why is the perplexity of CoCon (and PPLM) consistently lower than the perplexity of the baseline LM GPT-2? Shouldn't we expect a trade-off between controllability and text quality? In the PPLM paper, the perplexity of the baseline is consistently (slightly) lower than that of PPLM.

* Why does training on (human-written) Webtext instead of (machine-written) outputs of GPT-2 _decrease_ the text quality? Wouldn't we expect the opposite?

* The above three questions lead me to believe that using perplexity on GPT-1 might not be a suitable metric to judge text quality in these scenarios. Could you please provide more arguments why you believe a human study is not needed here?

Suggestions:

The model extensions listed under 4.4 are very interesting. The paper would be even stronger if you had quantitative experiments for these, as the examples that are given are not very convincing. For example, couldn't you apply your model to (gradual) sentiment transfer by conditioning CoCon on the input text as well as the target sentiment ("is perfect"), weighted by $\tau_{content}$? Even if the results were not very good compared to the state-of-the-art in sentiment transfer, such an experiment could show off the versatility of CoCon compared to PPLM. Moreover, if PPLM and CoCon complement each other as you claim, why not add another row "CoCon + PPLM" to Table 3, 4, and 5?

Minor suggestions:
* In the results section of 4.1, you say that L_adv marginally reduces CoCon's perplexity, but the table shows that _removing_ L_adv reduces it.

---

> ### Author Response · Authors · 2020-11-23
> **Response to Review 1 (Part 1)**
>
> We thank the reviewer for the positive and helpful comments. Please refer to the following for our response (in 2 parts):
>
> a) “The experimental section could be more thorough. For example, several aspects of the model are only evaluated qualitatively, and I don't find the examples very convincing. Moreover, some of the results are difficult to interprete or non-conclusive. The paper could benefit from a human evaluation.”
>
> We thank the reviewer for the helpful feedback. We are happy to share that we have gained approval to conduct the human evaluation (after the ICLR submission deadline) and have since added human evaluation results (Table 5 and 6) to better evaluate CoCon’s effectiveness in controlling high-level attributes such as sentiments and topics as well as fluency perceived by humans. The human evaluation corroborates the results from automatic evaluation where CoCon displays better control over topic- and sentiment-relevant generations than other controlled generation baselines, albeit with a slight tradeoff in fluency.
>
> b) “In the CoCon Setup you report to split … somewhere between the 8th and 12th BPE position. Why is this sufficient? Wouldn't we expect the model to perform poorly on prefixes that are not between 8 and 12 BPE tokens long?”
>
> We thank the reviewer for the discussion. We selected somewhere between 8th and 12th BPE position to strike a balance between learning convergence and generalization. On one hand, it would be easier for CoCon to learn to reconstruct the generation if the content input phase is very short, however, CoCon might not generalize well to longer content input during inference. On the other hand, using a long content input might make it challenging for CoCon to reconstruct the long generation faithfully at the start of the training, potentially causing an issue in training convergence. Indeed, we were surprised that CoCon is able to generalize well to both very short (one word for topic/sentiment control) and longer content input (GPT-2 passages in the new CoCon+ results) during inference, showing comparable human and automatic fluency scores in our experiments.
>
> c) “Table 2 suggests that CoCon without the adversarial loss achieves the best performance, drastically improving on content similarity while retaining comparable text quality and diversity. This makes me wonder why the adversarial term was introduced in the first place, and why it is apparently used in the other two experiments.”
>
> We included an adversarial loss in CoCon as it has been shown to improve fluency for text generation in prior work [1]. Since we have not been able to conduct a human evaluation before the initial submission due to approval issues, we included the adversarial loss in the other two experiments with speculation that it may benefit human-perceived fluency since the perplexity scores are close while automatic and human evaluations of fluency sometimes contradict each other. From our newly added human evaluation (Table 8 in the appendix of revision), we observe that humans do perceive CoCon without adversarial loss as more fluent, corroborating the findings from their perplexity score. We speculate that the addition of another adversarial loss to the existing set of other training objectives has a slightly counterproductive effect by making it more challenging for the CoCon model to converge in its training. We have included this discussion in the revised manuscript in the “Results” section of 4.1: “In our human evaluation (Table~8 of Appendix), we observe that humans also perceive CoCon without $\mathcal{L_{\text{adv}}}$ as more fluent, indicating that the addition of $\mathcal{L}_{\text{adv}}$ may have made it more challenging for the CoCon model to converge in its training.”
>
> d) “Why is the perplexity of CoCon (and PPLM) consistently lower than the perplexity of the baseline LM GPT-2? Shouldn't we expect a trade-off between controllability and text quality? In the PPLM paper, the perplexity of the baseline is consistently (slightly) lower than that of PPLM.”
>
> We thank the reviewer for bringing this up for discussion. We speculate that the perplexity of the baseline LM GPT-2 is lower as it is evaluated on the GPT model which has different model architecture and hence different bias compared to the GPT-2 model for generated tokens. In our newly added human evaluations (Table 5 of the revised manuscript), we observe that the baseline LM GPT-2 indeed has the highest fluency score across the topic/sentiment controlled generation, aligned with the trade-off between controllability and text quality we would expect.

---

> > ### Author Response · Authors · 2020-11-23
> > **Response to Review 1 (Part 2)**
> >
> >
> > e) “Why does training on (human-written) Webtext instead of (machine-written) outputs of GPT-2 decrease the text quality? Wouldn't we expect the opposite?”
> >
> > We have since conducted and added human evaluation of fluency to further investigate this observation. Training on (human-written) Webtext instead of (machine-written) outputs of GPT-2 also decreases the human-perceived fluency score slightly. We speculate that since the (machine-written) outputs of GPT-2 are generated by the LM itself, CoCon’s prediction can more easily match the training labels, hence helping CoCon to better converge during training, consequently generating texts of slightly higher quality.
> >
> >
> > f) “The above three questions lead me to believe that using perplexity on GPT-1 might not be a suitable metric to judge text quality in these scenarios. Could you please provide more arguments why you believe a human study is not needed here?”
> >
> > We thank the reviewer for the suggestion. We have since added human evaluation after gaining approval to conduct them recently.
> >
> > g) “The model extensions listed under 4.4 are very interesting. The paper would be even stronger if you had quantitative experiments for these..”
> >
> > We thank the reviewer for the helpful suggestion. We have since added more experiments to show more quantitative results for multiple content inputs by using GPT-2 output texts as an additional content input on top of topic/sentiment content inputs to show the flexibility of CoCon to control both high-level attributes and content of the generation. This dual content input generations from CoCon are labeled as CoCon+ in our revised manuscript. Table 6 and Table 7 show CoCon+ ‘s effectiveness in controlling the text’s content versus other baselines when evaluated with human and automatic metrics respectively.  The higher content similarity to the additional GPT-2 passage content input shows that CoCon+ can generate text that is more similar to generic content inputs (GPT-2 passage) than other controlled text generation methods which share similar prompt text and target attributes.
> >
> >
> > h) “For example, couldn't you apply your model to (gradual) sentiment transfer by conditioning CoCon on the input text as well as the target sentiment ("is perfect"), weighted by τcontent? Even if the results were not very good compared to the state-of-the-art in sentiment transfer, such an experiment could show off the versatility of CoCon compared to PPLM.”
> >
> > We thank the reviewer for the constructive comment. We have conducted additional experiments to study the suggested example in a similar spirit. In Table 6 where we compared CoCon with CoCon+ (described in response to (g)) where an additional content input is used, the influence from the original topic/sentiment content input is reduced as shown by CoCon+’s lower accuracy %. However, CoCon+’s topic/sentiment transfer is still present as shown in the higher CoCon+’s accuracy versus the un-conditioned GPT-2 baseline, also in Table 6. Moreover, we observe that CoCon+’s texts have higher content similarity to the additional GPT-2 passage content input, showing show off the versatility of CoCon to condition on multiple generic content inputs compared to PPLM.
> >
> > i) “Minor suggestions”
> >
> > We thank the review for point this typo and have corrected it accordingly.
> >
> > [1] Unsupervised text style transfer using language models as discriminators.  NeurIPS 18

---

### Official Review · AnonReviewer2 · 2020-10-28
**The paper proposed a novel way of controlling language model output.**

**Rating:** 7
**Confidence:** 3

**Review:**

The paper proposed a way to control the content output of a DNN-based language model (GPT-2 in the experiment, but not limited to it). It places an layer (CoCon) that can take an arbitrary phrase as the hint after generating the embedding but before generating the text. Experiments showed that the control is effective at directing the generated text. Examples confirmed that too.

Quality:

The design of the CoCon layer is intuitive. The authors clearly explained the rationale behind the design of the layer. Experiments are based on strong baseline (GPT-2, PPLM and CTRL), and show clear advantage of the model.

Clarity:

The writing is clear and easy to follow. I have some minor comments but believe they are easily fixable.

Originality:

CoCon has clear but incremental difference than PPLM and CTRL.

Significance:

Controlling the generation of LM is not a novel task. This is an improvement on an existing problem with several solutions. Moderate originality.

My questions and suggestions:

1) Page 2, core  contribution, item 3: what does "competitively" mean here?

2) Page 2, Related Work, first paragraph. "our approach aims to control the generation at a content level, beyond high-level
text attributes." should it be "at the content level"?

3) Page 5. cycle reconstruction loss. It would be helpful to give an example, otherwise it's a bit hard to see how cycle recon could have helped.

Same line: "unlikely co-occurs" -> "unlikely to co-occur" ?

4) Page 6 , 2nd paragraph "self-supervised and requires no manually labeled data fully" is duplicated, can be removed.

5) Overall speaking, the choice of content input for all examples are weird. Why do we use partial phrases without a clear meaning or subject as the content hint?

---

> ### Author Response · Authors · 2020-11-23
> **Response to Review 2**
>
> We thank the reviewer for the positive comments and feedback. Please refer to the following for our response:
>
>
> a) “what does "competitively" mean here?”
>
> We thank the reviewer for point this out for clarification. We initially use “competitively” to mean that CoCon can outperform the baselines in most of our initial experiments. We have since changed the phasing of the core contribution to the following for more clarity:
>
> “Through ablation studies and comparisons with strong baselines like PPLM and CTRL, we investigate how CoCon effectively influences the content of generation and can competitively control high-level text attributes such as topic and sentiment.”
>
>
> b) “Page 5. cycle reconstruction loss. It would be helpful to give an example, otherwise it's a bit hard to see how cycle recon could have helped.”
>
> We thank the reviewer for the constructive comment. We have since added Figure 2 with an example to improve the understanding of cycle reconstruction loss and better contrast it with self reconstruction.
>
> c) “Overall speaking, the choice of content input for all examples are weird. Why do we use partial phrases without a clear meaning or subject as the content hint?”
>
> We used partial phrases for the experiments in “Section 4.1: Content Similarity” to study how CoCon can condition on generic content input in a large scale manner. For topic and sentence control (Section 4.2 and 4.3), the content inputs are control code words and sentiment markers used in previous methods respectively.
>
> In our newly added experiments and results, we use GPT-2 output texts (instead of partial phases) as the second content input on top of the topic/sentiment content input to better study how CoCon can generate text of content similarity of unseen content input, marked as CoCon+ in the revised manuscript. Table 6 and Table 7 show CoCon+ ‘s content similarity with the conditioning GPT-2 text when evaluated with human and automatic metrics respectively.  The higher content similarity to the additional GPT-2 passage content input shows that CoCon+ can flexibly generate text that is more similar to generic content inputs (GPT-2 passage) than other controlled text generation methods which share similar prompt text and target attributes.
>
>
> d) “should it be "at the content level"?”
>
> We thank the reviewer for point this out. Indeed, that would be better. We have edited accordingly.
>
> e) "unlikely co-occurs" -> "unlikely to co-occur"
>
> We thank the reviewer for the suggestion and have edited it accordingly.

---

### Official Review · AnonReviewer4 · 2020-10-28
**Review: CoCon: A Self-Supervised Approach for Controlled Text Generation**

**Rating:** 6
**Confidence:** 4

**Review:**

SUMMARY:

The paper proposes a self-supervised technique for controlling the productions of a Transformer-based pretrained generator. The technique consists in augmenting the architecture of the pretrained model with a special "content-conditioner" (CoCon) block which is able to exploit a contextual condition.
At training time, this contextual condition is obtained, in a self-supervised way, by removing a textual portion from a training text and using this portion as the contextual condition, and then the parameters of the CoCon component learn how to approximately recover the missing portion based on this context (the portion itself) and on the prefix text preceding the removal.
At test time, a textual condition is provided as context, and the trained model produces a text "imbued" (authors' terminology = influenced) by this condition.

POSITIVES:

While self-supervised learning has been employed for certain text generation tasks, such as summarization, I am not aware of previous works directly concerned with self-supervision for controlled open-ended text generation. This appears to be a very worthwhile direction to pursue.

ISSUES and QUESTIONS:

*Clarity*. The main idea is actually pretty simple but the reader has to wait until the end of page 4 (Self Reconstruction Loss) to be able to understand it (true: it was exposed in the intro, but in a way difficult to understand on a first reading), and is a bit drowned in a dense mass of mathematical notations that do not help.  Some parts of the formal description are quite difficult to follow, for instance the section on "Cycle Reconstruction Loss". Also, in Fig. 1, the reader does not immediately see that (*IF* I understand correctly) the hidden states $h_{t-1},...,h_{l-1}$ are masked, which does not help in understanding an already dense formal description. Perhaps most serious: the central objective of a text "imbued" (i.e. "influenced") by a conditioning text is left pretty informal.

*Related work and Alternatives to the CoCon block*. Adapter Layers (https://www.aclweb.org/anthology/D19-1165/) are a technique for adapting pretrained models which does not require retraining the entire model; they are therefore similar in spirit to the CoCon block, and *should* be cited, with differences highlighted. (More minor: it might (?) also be worthwhile to mention a different option: using an encoder-decoder model (similar to NMT) where the conditioning context would just be the "source" and the generated text the "target", directly providing an attention-driven mechanism --- however the issue of retraining the whole model would then need to be addressed)).

*Complexity of the overall model, intuition about the different losses, hyperparameters* The self-reconstruction loss, by itself, appears to be problematic. Indeed, a model trained only on this loss might just learn to *copy* the conditioning text, thus destroying fluency and generalization. This should be explicitely discussed (instead of leaving the point more or less implicit in the second paragraph of section 4.4: "... limit where the texy appears incomprehensible"). Therefore the need to interpolate this loss with other losses (section 3.1). While you provide some ablation experiments, you do not much discuss the importance of these different losses. In particular, you should give more intuition/motivation for the Cycle Reconstruction Loss, which I did not really understand. The overall model involves quite a few hyperparameters ($\lambda$'s in equation (20), $\tau_{content}$.

*Results* The results are difficult to interpret, in particular due to the not very clearly formalized control objective (do you want the generated text to contain literal parts of the conditioning text (apparently not), or to have some semantic simlarity with the conditioning text (apparently yes, but you do not explicitly mention or define semantic similarity)? It is difficult for the reader to really assess the quality of the results. Here, a human evaluation with a clear evaluation protocol would really be useful.


Overall, an interesting and important objective: self-supervision of controlled text generation, with some nice ideas. But serous flaws in presentation and experimental validation.

-------
**Written after rebuttal:**
Thank you for the substantial improvements to the paper in terms of clarity (in particular Figure 2 is helpful) and additional experiments/human evaluations. Despite some underlying questions (from me and other reviewers) that remain, I have updated my score and am now leaning towards acceptance.

---

> ### Author Response · Authors · 2020-11-23
> **Response to Review 4 (Part 1)**
>
> We thank the reviewer for the constructive and detailed comments. Please refer to the following for our response (in 2 parts):
>
> a) “The main idea is actually pretty simple but the reader has to wait until the end of page 4 (Self Reconstruction Loss) to be able to understand it”
>
> We thank the reviewer for the helpful feedback. We have edited the following sentence in the introduction to better introduce CoCon’s self reconstruction loss earlier on:
>
> Original: “By splitting each text sequence into two segments ($[\mathbf{x}^a ; \mathbf{x}^b]$), CoCon learns to help the LM reconstruct missing latter segments ($\mathbf{x}^b$) by taking $\mathbf{x}^b$ itself as the content input. “
> --->
> Revised: “By splitting each text sequence into two segments ($[\mathbf{x}^a ; \mathbf{x}^b]$), CoCon learns through a self reconstruction objective to help the LM reconstruct missing latter segments ($\mathbf{x}^b$) by taking $\mathbf{x}^b$ itself as the content input. “
>
> We have separated the model architecture and training objectives of CoCon for the reader to more easily understand these two components of CoCon. To improve the reader’s understanding of CoCon, we have also added Figure 2 in the revision to better illustrate the two main CoCon training objectives: self reconstruction and cycle reconstruction.
>
> b) “in Fig. 1, the reader does not immediately see that (IF I understand correctly) the hidden states
> Ht−1,...,hl−1 are masked”
>
> We thank the reviewer for bringing this up for clarification. The mask is only applied for the self reconstruction training objective on the $\mathbf{c}$ according to which token CoCon is predicting. Since $\mathbf{x}^b$ is used as $\mathbf{c}$ during the self reconstruction training, CoCon may trivially copy the whole $\mathbf{c}$ to predict $\mathbf{x}^b$ rather than learning to generate text fluently. At the step where CoCon is reconstructing $x_{i+1}$, only one token, $x_{i+1}$ of $\mathbf{c}$, is masked to avoid CoCon from trivially copying the exact token. This helps CoCon to learn to generate the token from other contexts such as prompt and content input text. We have edited the following sentence from Section “Self Reconstruction Loss” in \S 3.1 to improve clarity:
>
> Original: “To avoid trivializing the prediction of the next token $x_{i+1}$, we apply a self-token $\mathbf{c}$-mask at CoCon's attention layer such that $\mathbf{h_{i}}'$ does not attend to values computed from $\mathbf{h_{i+1}}$.”
> -->
> Revised: “To avoid trivializing the prediction of the next token $x_{i+1}$ during training, we apply a self-token $\mathbf{c}$-mask at CoCon's attention layer such that $\mathbf{h_{i}}'$ does not attend to the token $x_{i+1}$ in $\mathbf{c}$ it is trying to predict.”
>
> c) “Some parts of the formal description are quite difficult to follow, for instance, the section on "Cycle Reconstruction Loss".”
> We thank the reviewer for the constructive comment. We have since added Figure 2 with an example to improve the understanding of cycle reconstruction loss and better contrast it with self reconstruction.
>
>
>
> d) “the central objective of a text "imbued" (i.e. "influenced") by a conditioning text is left pretty informal.”
>
> We thank the reviewer for pointing this out. We have replaced the word “imbued” to “conditioned on” which is more oftenly used in the literature.
>
> e) “Adapter Layers are a technique for adapting pretrained models which does not require retraining the entire model; they are therefore similar in spirit to the CoCon block, and should be cited”
>
> We thank the reviewer for bringing this relevant work up for discussion. While adapter layers [1] have been previously proposed to also save on model size and training resources for multilingual translation its training differs from CoCon’s self-supervised learning in that it relies on supervised training for a different task of machine translation, using annotated sentence pairs of different languages. CoCon’s core contribution is the use of self-supervised learning objectives such as self and cycle reconstruction to facilitate its training for conditioned text generation. We have added the following sentence in the “Related Work” section to cite the work and discuss its difference from CoCon:
>
> “Small adapter layers [1] have been previously proposed for multilingual translation to also save on model size and training resources but differ from CoCon's self-supervised training as they rely on annotated sentence pairs of different languages for supervised training.”

---

> > ### Author Response · Authors · 2020-11-23
> > **Response to Review 4 (Part 2)**
> >
> >
> > f) “The self-reconstruction loss, by itself, appears to be problematic. Indeed, a model trained only on this loss might just learn to copy the conditioning text, thus destroying fluency and generalization. This should be explicitly discussed (instead of leaving the point more or less implicit in the second paragraph of section 4.4”
> >
> > We thank the reviewer for the helpful suggestion. We have since edited and included the following sentences (in Section “Self Reconstruction Loss” in Section 3.1) to better discuss this issue and how it is been addressed in CoCon’s $\mathbf{c}$-mask in self reconstruction training to avoiding learning to copy text directly:  “To avoid trivializing the prediction of the next token $x_{i+1}$ during training, we apply a self-token $\mathbf{c}$-mask at CoCon's attention layer such that $\mathbf{h}'_{i}$ does not attend to the token $x_{i+1}$ in $\mathbf{c}$ it is trying to predict.”
> >
> > We also make it more explicit, in “Cycle Reconstruction Loss” paragraph of Section 3.1, that cycle reconstruction loss is proposed for CoCon to generalize when $\mathbf{c}$ is different from the generation label: “The self reconstruction loss relies on CoCon content input ($\mathbf{c}$) and initial prompt text ($\mathbf{p}$) originating from one single text sample. To encourage generalization on cases where $\mathbf{c}$ and $\mathbf{p}$ are from divergent text sources, we employ a cycle reconstruction training that utilizes two different training samples”
> >
> > g) “In particular, you should give more intuition/motivation for the Cycle Reconstruction Loss, which I did not really understand.”
> >
> > We thank the reviewer for the constructive comment. We have since added Figure 2 with an example to improve the understanding of cycle reconstruction loss and better contrast it with self reconstruction.
> >
> >
> > h) “The results are difficult to interpret, in particular due to the not very clearly formalized control objective (do you want the generated text to contain literal parts of the conditioning text (apparently not), or to have some semantic similarity with the conditioning text (apparently yes, but you do not explicitly mention or define semantic similarity)? It is difficult for the reader to really assess the quality of the results. Here, a human evaluation with a clear evaluation protocol would really be useful.”
> >
> > We thank the reviewer for the helpful suggestion. We have since added human evaluation on the semantic similarity with the conditioning text. Indicated as CoCon+ in the revised manuscript, on top of the (first) target topic/sentiment content input, we also condition CoCon on GPT-2 output text as the second content input. These GPT-2 output texts are generated from the same prompt text as CoCon and the other baselines. Table 6 and Table 7 show CoCon+ ‘s content similarity with the conditioning GPT-2 text when evaluated with human and automatic metrics respectively.  The higher content similarity to the additional GPT-2 passage content input shows that CoCon+ can flexibly generate text that is more similar to generic content inputs (GPT-2 passage) than other controlled text generation methods which share similar prompt text and target attributes.
> >
> >
> > [1] Ankur Bapna, Naveen Arivazhagan, and Orhan Firat. Simple, scalable adaptation for neural machine translation. EMNLP-IJCNLP 2019

---

### Official Review · AnonReviewer3 · 2020-10-28
**An architectural modification allowing integrating of textual contexts, converting the problem of controlled text generation into conditional text Generation.**

**Rating:** 4
**Confidence:** 5

**Review:**

This paper tackles the problem of controlled text generation by converting it into a conditional text generation similar to (Keskar et al.19).  It proposes an architectural modification to the transformer LM used in GPT2,  Specifically, a CoCon layer is added as a separate transformer block in the middle allowing self-attention to be performed between the textual context representations LM_α(c) and the generations LM_α(x_{:t-1}) this is performed through concatenating the key and value matrices with the keys and values of the encoded textual context. Authors provide 4 different losses to train this additional layer.

Pros:
-  The proposed method has an advantage over (Keskar et al.19) by
1) avoiding rigid control tokens and replacing them by textual context.
2) avoiding to retrain the whole LM architecture and replacing this by retraining single transformer block instead
3) allowing several control contexts at once (this is an interesting aspect of the proposed solution)

Cons:
- The proposed solution to controlled NLG is simple yet not inspiring nor revolutionary, simplicity could have been an advantage here ofc, if it tackled the problem providing a concrete method to control NLG models. however, this is not the case here (See next)

- Conditioning NLG models on textual contexts to influence the generated text is a straight forward solution and makes sense in terms of flexibility, and in fact, has been used before [1] to enhance faithfulness in QG tasks. On the other hand, conditional text generation as a solution to controlled NLG might be effective for influencing topic or sentiment, however, this formulation is not suitable for other types of control where textual contexts are hard to formulate, such as "removing" toxicity, controlling the length.

- There might be an issue with the Adversarial loss, being non-differentiable (see Q1 below)

- Evaluation could have been more thorough, specifically, when the proposed method has a superior topic and sentiment relevance in table 3 and table 4 this comes on the cost of perplexity. Enhancing topic relevance of PPLMand CTRL could be achieved by reducing the temperature during decoding, on the expense of Perplexity as well. This is similar to the Quality/Diversity tradeoff showed in [2]. While this has been an issue in previous work as well, it would have been better to fix this issue and provide as better evaluation, a good method to evaluate this could have been plotting perplexity vs control satisfaction rate under a temperature sweep.

- There are missing details on how the textual contexts are selected during inference time. In most of the cases, they're handcrafted topic names or short sentences "is perfect". This makes the proposed solution very similar to control tokens by Keskar et al. 19. One advantage of using textual control tokens is handling unseen "content inputs" at test time. This should have been evaluated to show the superiority of this solution.

Questions:

Q1: This is a critical one, If I got this part correctly, the adversarial loss eq 18, requires to sample y from the LM, this is non-differentiable. if that is the case, did you follow any necessary steps (e.g. RL or continuous approx) to overcome this non-differentiability?


refs:

1- Zero-Shot Question Generation from Knowledge Graphs for Unseen Predicates and Entity Types, NAACL2018

2- LANGUAGE GANS FALLING SHORT ICLR2020

minor:
- Is that a typo in figure 1? The cocon layer output representation should be h_1 , h_{t-2} should be h' _ 1 , h' _ {t-2}

---

> ### Author Response · Authors · 2020-11-23
> **Response to Review 3 (Part 1)**
>
> We thank the reviewer for the thoughtful and helpful comments. Please refer to the following for our response (in 2 parts):
>
> a) “Conditioning NLG models on textual contexts to influence the generated text is a straight forward solution and makes sense in terms of flexibility, and in fact, has been used before [1] to enhance faithfulness in QG tasks.”:
>
> While conditioning neural language generation on textual context has been used before, CoCon is the first to learn zero-shot conditioned language generation for large language models (LMs) in a self-supervised manner. [1] enhances faithfulness in question generation by attending to textual context such as predicates, subject types or object types rather than the content input used here in CoCon. We have added this discussion in Section 2 “Related Work” of the revised manuscript. Given how remarkable current larget transformer LMs are in text generation, we believe it is timely that CoCon can extend the LMs’ potential to even more applications through better control of its generation.
>
>
> b) “this formulation is not suitable for other types of control where textual contexts are hard to formulate, such as "removing" toxicity, controlling the length.”
>
> Indeed. However, CoCon’s main aim is to exercise fine-grained control over LM’s generations with the flexible medium of content input. This makes CoCon a complementary and orthogonal tool to other types of controlled generation methods as shown in ‘Complementary Text Control’ of Section 4.4.
>
> c) “There might be an issue with the Adversarial loss, being non-differentiable (see Q1 below)”
>
> We thank the reviewer for point this out for clarification. Similar to previous work on adversarial learning for text generation [2], through continuous approximation of discrete sampling of $y$, CoCon and $f_{\text{disc}}$ can be trained with backpropagation in an end-to-end manner. We have added this detail in our revision under “Adversarial Loss” in Section 3.1:
> “Through continuous approximation of discrete sampling of $y$, CoCon and $f_{\text{disc}}$ can be trained with backpropagation in an end-to-end manner.”
>
>
> d) “Evaluation could have been more thorough”
> We thank the reviewer for the constructive suggestion. We have since added more results to show CoCon’s superiority in controlling the generation with unseen "content inputs" as shown in Table 6 and Table 7. Please refer to the detailed discussion in the response (g) below.
>
> We have also added human evaluation (Table 5 and 6) to better evaluate CoCon’s effectiveness in controlling high-level attributes such as sentiments and topics as well as fluency perceived by humans. The human evaluation corroborates the results from automatic evaluation where CoCon displays better control over topic- and sentiment-relevant generations than other controlled generation baselines, albeit with a slight tradeoff in fluency.
>
>
> e) “Enhancing topic relevance of PPLM and CTRL could be achieved by reducing the temperature during decoding, on the expense of Perplexity as well … While this has been an issue in previous work as well, it would have been better to fix this issue and provide as better evaluation, a good method to evaluate this could have been plotting perplexity vs control satisfaction rate under a temperature sweep.”
>
> We thank the reviewer for the suggestion. We have added more results that could better address this point. When adding an additional (second) content input to CoCon on top of the target topic/sentiment (first) content input (named CoCon+ in the revised manuscript), we observe a tradeoff in the topic/sentiment conditioned generation (CoCon vs CoCon+ in Table 6). While CoCon+ generation shows higher content similar to the additional (second) content input than CoCon, its generations displayed lower topic/sentiment relevance. We have edited the third core contribution in the Introduction to better reflect this point without claiming that CoCon outperforms PPLM and CTRL in topic/sentiment control: “Through ablation studies and comparisons with strong baselines like PPLM and CTRL, we investigate how CoCon controls high-level attributes such as topic and sentiment while generating texts that have high content similarity to conditioning text.”
>
> We would like to also point out that while baseline methods like PPLM and CTRL can also control high-level attributes like topics and sentiment, CoCon offers additional fine-grained control and flexibility over text generations by conditioning on unseen "content inputs", as discussed further in response (g) below. Moreover, CoCon can be trained in a self-supervised manner, relieving the burden of data annotation involved in those previous methods.

---

> > ### Author Response · Authors · 2020-11-23
> > **Response to Review 3 (Part 2)**
> >
> > f) “There are missing details on how the textual contexts are selected during inference time.”
> >
> > We thank the reviewer for bringing this up for clarification. The CoCon content inputs 'is perfect' and 'is horrible' are positive and negative sentiment attribute markers [3]. Sentiment attribute markers are essentially n-grams that appear in high frequency in text samples annotated with a particular attribute such as positive/negative sentiment. While we use one sentiment marker each for evaluation in the main paper, we also included generation from other positive/negative sentiment markers in Table 12 for more examples. The topic content inputs   'computers', 'politician', 'religion' and 'scientist' mirrors the CTRL’s control codes [4]. We have added these details in the revision’s “Setup” in Section 4.3:
> >
> > “Sentiment attribute markers [3] 'is perfect' and 'is horrible' are used as content inputs to generated CoCon outputs for the Positive and Negative sentiment respectively. Sentiment attribute markers are n-grams that appear in high frequency in text samples annotated with a particular attribute such as positive/negative sentiment.”
> >
> >
> >
> > g) “One advantage of using textual control tokens is handling unseen "content inputs" at test time. This should have been evaluated to show the superiority of this solution.”
> >
> > We thank the reviewer for the helpful suggestion. We have since added more experiments to show the superiority of CoCon in controlling the text generation to unseen “content inputs” by using GPT-2 output text as the content input on top of topic/sentiment content inputs to show the flexibility of CoCon to control both high-level attributes and content of the generation. This dual content input generations from CoCon are labeled as CoCon+ in our revised manuscript. Table 6 and Table 7 show CoCon+ ‘s effectiveness in controlling the text’s content versus other baselines when evaluated with human and automatic metrics respectively.  The higher content similarity to the additional GPT-2 passage content input shows that CoCon+ can generate text that is more similar to unseen content inputs (GPT-2 passage) than other controlled text generation methods even though these methods share similar prompt text and target attributes.
> >
> > h) “Is that a typo in figure 1? The cocon layer output representation should be ..”
> >
> > We thank the reviewer for point this out for clarification. That is not a typo: $\tilde{\mathbf{o_t}}$ from Equation (7),  represents the logit of the first CoCon generated token where the prompt text’s original hidden state ($\mathbf{h_{:t-2}}$) is concatenated to the CoCon state $\mathbf{h_{t-1}}'$. The original hidden states of the prompt text are used as the prompt text is not an output of the CoCon.
> >
> >
> >
> > We would like to thank the reviewer again for the very helpful and thoughtful feedback for us to improve the paper.
> >
> > [1] Zero-Shot Question Generation from Knowledge Graphs for Unseen Predicates and Entity Types, NAACL2018
> > [2] Unsupervised text style transfer using language models as discriminators.  NeurIPS 18
> > [3]    Delete, retrieve, generate:  A simple approach to sentiment and style transfer. NAACL 18
> > [4] Ctrl:   A  conditional  transformer  language  model  for  controllable  generation.

---

### Author Response · Authors · 2020-11-23
**Overall Response to All Reviewers**

We would like to thank all the reviewers for the insightful and valuable comments. We have revised our paper based on the comments and provided the individual response to each reviewer. A summary of the key revision is presented here:

+ Added human evaluation results for topic/sentiment relevance and fluency of CoCon and baselines’ text generation (Table 5 & 6).

+ Added Figure 2 to better explain CoCon’s cycle reconstruction loss.

+ Added human and automatic evaluation to better study CoCon’s versatility of conditioning on unseen content input on top of topic/sentiment control (Table 6 & 7)

+ Edited Figure 1 and Section 3 to improve clarity.

+ Incorporated discussions and clarification pointed out by reviewers in the main text body of the manuscript.

---

### Decision · Program_Chairs · 2021-01-07
**Final Decision**

**Decision:**

Accept (Poster)

**Comment:**


The paper aims at controllable generation by introducing an additional "content-conditioner" block in the Transformer models. The paper further provides 4 different variants of a pre-training task to train the content-conditioner model.

While the proposed approach seems an incremental contribution over CTRL and PPLM, certain reviews praised the approach being novel while keeping the architecture changes minimal. Overall, reviews indicate that the overall proposed method of fine-grained controlled generation with self-supervision is valuable, and empirical results support its effectiveness.

All reviewers initially raised concerns regarding clarity and lack of human evaluation. However, clarity issues seem to be resolved through author/reviewer discussions and the updated revision.

R3 had important concerns regarding topic and sentiment relevance evaluations.
While the reviewer remains unconvinced after discussions with authors, after carefully reading the revised paper and discussions, I feel that the authors tried to address this point fairly  through their additional experiments and also edited their contribution statement accordingly.

Overall, at least two reviewers sounded very excited about this work and other than R3's concerns, the general sentiment about this work was positive. Therefore, I recommend weak accept.

There are still some writing issues that I strongly encourage authors to carefully address in the future versions. Quoting from reviewer discussions:

> Differentiability of the adversarial loss. Authors just added one statement saying " Through continuous approximation.." without any more details are given, which continuous approx was used (Gumbel softmax?) and how they overcame the problem of its training instability.

> Table 6, can be misleading, authors bold the results when cocon+ is performing better than baselines (mostly in content similarity) but not the other way around topic/sentiment accuracy. The latter is arguably more important.